# Spleen tyrosine kinase mediates innate and adaptive immune crosstalk in SARS-CoV-2 mRNA vaccination

Sebastian J Theobald[1,2,†], Alexander Simonis[1,2,†] (iD), Julie M Mudler[1,2,†] (iD), Ulrike Göbel[3],
Richard Acton[3] (iD), Viktoria Kohlhas[1,2,3] (iD), Marie-Christine Albert[3,4], Anna-Maria Hellmann[2,5],
Jakob J Malin[1,2] (iD), Sandra Winter[1,2], Michael Hallek[1,2,3], Henning Walczak[3,4,6] (iD),
Phuong-Hien Nguyen[1,2,3] (iD), Manuel Koch[2,4,7] (iD) & Jan Rybniker[1,2,8,*] (iD)

## Abstract

**Durable cell-mediated immune responses require efficient innate immune signaling and the release of pro-inflammatory cytokines. How precisely mRNA vaccines trigger innate immune cells for shaping antigen specific adaptive immunity remains unknown. Here, we show that SARS-CoV-2 mRNA vaccination primes human monocyte-derived macrophages for activation of the NLRP3 inflammasome. Spike protein exposed macrophages undergo NLRP3-driven pyroptotic cell death and subsequently secrete mature interleukin-1β. These effects depend on activation of spleen tyrosine kinase (SYK) coupled to C-type lectin receptors. Using autologous cocultures, we show that SYK and NLRP3 orchestrate macrophage-driven activation of effector memory T cells. Furthermore, vaccination-induced macrophage priming can be enhanced with repetitive antigen exposure providing a rationale for prime-boost concepts to augment innate immune signaling in SARS-CoV-2 vaccination. Collectively, these findings identify SYK as a regulatory node capable of differentiating between primed and unprimed macrophages, which modulate spike protein-specific T cell responses.**

**Keywords** inflammasome; innate immunity; mRNA vaccines; SARS-CoV-2; SYK signaling

**Subject Categories** Immunology; Microbiology, Virology & Host Pathogen Interaction

**EMBO Mol Med (2022) e15888**

## Introduction

SARS-CoV-2 mRNA vaccines play a key role in controlling the ongoing coronavirus disease 2019 (COVID-19) pandemic. Understanding human immune responses to these novel vaccine constructs, which express the SARS-CoV-2 spike protein (S-protein), is essential for multiple reasons. While the knowledge gap is increasingly addressed with regard to the adaptive immune system involving S-protein-specific T and B cell responses (Goel *et al*, 2021; Turner *et al*, 2021), little is known about innate immune cell signaling and crosstalk between innate and adaptive immunity following vaccination. Durable pathogen specific immune responses require innate immune cell-activation via adjuvants or pathogen associated molecular patterns (PAMP). Especially the activation of the NOD-, LRR- and pyrin domain-containing protein 3 (NLRP3) inflammasome in macrophages and dendritic cells with subsequent secretion of interleukin-1β (IL-1β) seems to be crucial for an efficient cell-mediated and humoral immune response (Ichinohe *et al*, 2009; Evavold & Kagan, 2018; Munoz-Wolf & Lavelle, 2018; Van Den Eeckhout *et al*, 2020; Hatscher *et al*, 2021). In addition, inflammasome formation and IL-1β secretion through activation of Toll-like receptors are key features of classical vaccine adjuvants (Pulendran *et al*, 2021).

NLRP3-dependent secretion of active IL-1β is a highly controlled, multi-step process requiring a priming signal in which cellular receptors recognize conserved PAMPs leading to NLRP3 and pro-IL-1β expression. A second activation step is triggered by a range of intrinsic or pathogen derived stimuli such as adenosine triphosphate (ATP), microbial toxins such as nigericin or nucleic acids. This step leads to oligomerization of the adaptor protein ASC and NLRP3,

1 Department I of Internal Medicine, Faculty of Medicine, University of Cologne, Cologne, Germany
2 Center for Molecular Medicine Cologne (CMMC), Faculty of Medicine, University of Cologne, Cologne, Germany
3 Excellence Cluster on Cellular Stress Responses in Aging-Associated Diseases (CECAD), University of Cologne, Cologne, Germany
4 Center for Biochemistry, Faculty of Medicine, University Hospital of Cologne, Cologne, Germany
5 Department of Experimental Pediatric Oncology, University Children's Hospital of Cologne, Medical Faculty of Medicine, University Hospital of Cologne, Cologne, Germany
6 Centre for Cell Death, Cancer, and Inflammation (CCCI), UCL Cancer Institute, University College London, London, UK
7 Institute for Dental Research and Oral Musculoskeletal Biology, Medical Faculty, University of Cologne, Cologne, Germany
8 German Center for Infection Research (DZIF), Partner Site Bonn-Cologne, Cologne, Germany
*Corresponding author. Tel: +49 21 478 89611; E-mail: jan.rybniker@uk-koeln.de
†These authors contributed equally to this work

serving as an activation platform for caspase-1. Active caspase-1 in turn cleaves pro-IL-1β and gasdermin D (GSDMD), leading to secretion of mature IL-1β. Activated GSDMD forms membrane pores causing pyroptosis, a highly pro-inflammatory form of necrotic cell death (Mariathasan *et al*, 2006; Eisenbarth *et al*, 2008; Lamkanfi & Dixit, 2014; Broz & Dixit, 2016; Swanson *et al*, 2019). Recent data also highlight the importance of inflammasome activation in immunopathogenesis of COVID-19 (Theobald *et al*, 2021; Junqueira *et al*, 2022; Sefik *et al*, 2022; Zeng *et al*, 2022). There is evidence that the S-protein itself functions as a PAMP and driver of inflammatory cytokine signaling as shown in COVID-19 patient-derived macrophages (Lu *et al*, 2021; Theobald *et al*, 2021; Paludan & Mogensen, 2022). Since mRNA vaccines encoding the S-protein lack conventional vaccine adjuvants, it is currently unknown how pattern recognition receptors and cell death pathways are engaged in innate immune signaling of SARS-CoV-2 vaccinated individuals. Here, we investigated mechanisms and consequences of inflammasome activation in monocyte-derived human macrophages prior to and after prime-boost SARS-CoV-2 mRNA vaccination.

## Results

### Vaccination primes the NLRP3 inflammasome in macrophages

To profile macrophage activation in vaccinated individuals, we first isolated peripheral blood mononuclear cells (PBMCs) prior to (T0, $N = 44$) and 14 days after the first vaccination (T1, $N = 43$) as well as 14 days (T2, $N = 54$) and 70 days (T3, $N = 42$) after the second vaccination (Fig 1A). Healthy mRNA vaccinated individuals from the ISAVak cohort provided blood for this study (Table 1). CD14[+] monocytes were selected and differentiated to macrophages, followed by the exposure to affinity purified prefusion-stabilized S-protein (inflammasome priming) and nigericin or ATP for full NLRP3 activation (Fig 1A). While unstimulated and stimulated macrophages isolated prior to vaccination (T0) failed to secrete IL-1β, we were able to detect IL-1β in supernatants of macrophages at T1 (Figs 1B and C, and EV1A). Interestingly, macrophages of individuals having received a booster vaccination secreted IL-1β to significantly higher levels at T2 (Figs 1C and EV1A). At T3, 70 days after the second vaccination, macrophages remained responsive

toward S-protein priming and IL-1β levels were comparable to T1 values (Figs 1C and EV1A). Both ATP and nigericin were functional as inflammasome-activating signals in S-protein-primed post-vaccination macrophages (Figs 1C and EV1B). When testing LPS as an alternative NLRP3 stimulating PAMP, we observed similar levels of secreted IL-1β throughout all time-points indicating a certain S-protein-specific responsiveness induced in macrophages by SARS-CoV-2 mRNA vaccination (Figs 1D and EV1A). To verify functional adaptive immunity in vaccinated individuals of our cohort, we quantified S-protein-specific IgG plasma levels by ELISA at T0, T1, and T2, which correlated with increasing levels of IL-1β identified in supernatants of stimulated macrophages (Fig 1E). NLRP3 inflammasome activation in macrophages isolated from vaccinated individuals was confirmed by visualization of ASC oligomerization (ASC specks) using immunofluorescence microscopy. ASC specks were detectable in S-protein primed T2 macrophages but not in T0 macrophages (Fig 1F–H). Accordingly, we detected the cleaved substrates of NLRP3-activated caspase-1 (IL-1β and GSDMD-N) in S-protein-stimulated T2 macrophages only (Figs 1I and EV1C and D). LPS stimulation was used as a positive control (Fig 1I). Pyroptotic cell death concomitant with NLRP3 inflammasome activation could be confirmed by flow cytometry (Fig 1J and K) and by LDH release measured in supernatants of S-protein stimulated macrophages (Fig 1L). Treatment with MCC950, a selective NLRP3 inflammasome inhibitor, abrogated IL-1β release from pyroptotic cells (Fig 1M).

### IL-1β is upregulated in macrophages following vaccination

To mechanistically address the large differences observed in T0 and T2 macrophages, we first performed transcriptome analyses using RNA-seq. Data were derived from macrophages without any *ex vivo* stimulation. Gene set enrichment analyses showed that vaccination impacts on expression levels of genes associated with the immune response to pathogens and cytokine signaling (Fig 2A and B and Table EV1). IL-1β-specific mRNA was strongly upregulated in macrophages from vaccinated individuals. Since upregulation of IL-1β is a prerequisite but not the main trigger for secretion of the mature form of this cytokine, we speculated that vaccine-induced positive regulation of immune receptors and posttranscriptional modification of associated regulatory proteins drives assembly of the NLRP3 inflammasome following exposure to the S-protein.

**Figure 1. S-protein (SP) priming induces NLRP3 inflammasome activation in post-vaccination macrophages.**

A–D Monocytes were isolated prior (T0; *n* = 35 individuals) or after vaccination [2 weeks after 1st (T1; *n* = 28 individuals), 2 weeks (T2; *n* = 44 individuals) and 10 weeks (T3; *n* = 31 individuals) after 2nd vaccination] and differentiated to macrophages (A). IL-1β concentrations in supernatants of macrophages, unstimulated (control) (B) or stimulated with SP/N (C) or LPS/N (D).

E SARS-CoV-2 spike ELISA of the corresponding plasma: (EC50 titer) of samples collected at T0 (*n* = 35 individuals), T1 (*n* = 28 individuals) or T2 (*n* = 44 individuals).

F Immunofluorescence microscopy images of ASC specks (green) in T2 macrophages. Nuclei were stained with DAPI (blue). Scale bar indicates 20 μm. Arrow indicates ASC speck formation with magnified illustration in the right corner of the image.

G, H ASC specks were quantified microscopically at T0 (G) (*n* = 7 individuals) and T2 (H) (*n* = 8 individuals) upon indicated stimuli.

I Cleaved gasdermin D (lysate) and cleaved IL-1β (supernatant) detected by Western Blot in macrophages stimulated with LPS/N or SP/N at T0 and T2.

J, K Cell death quantification of macrophages by flow cytometry in unstimulated or SP/N stimulated cells at T0 (*n* = 5 individuals) (J) and T2 (*n* = 5 individuals) (K) using the viability dye Zombie UV. LPS/N was used as positive control.

L LDH release of macrophages (*n* = 6 individuals) stimulated with SP/N at T2 compared to unstimulated cells.

M IL-1β concentrations of MCC950 treated T2 macrophage (*n* = 44 individuals) stimulation with SP/N.

Data information: For statistical analysis, one-way ANOVA with Tukey *post-hoc* test was used. Box plots indicate the median and the upper and lower quartile. Outliers are plotted as individual dots (outside the 10–90 percentile). Scatter dot plots show mean ± SD. *P < 0.05; **P < 0.01, ***P < 0.001. (S-protein: SP, lipopolysaccharide: LPS, nigericin: N).

Source data are available online for this figure.

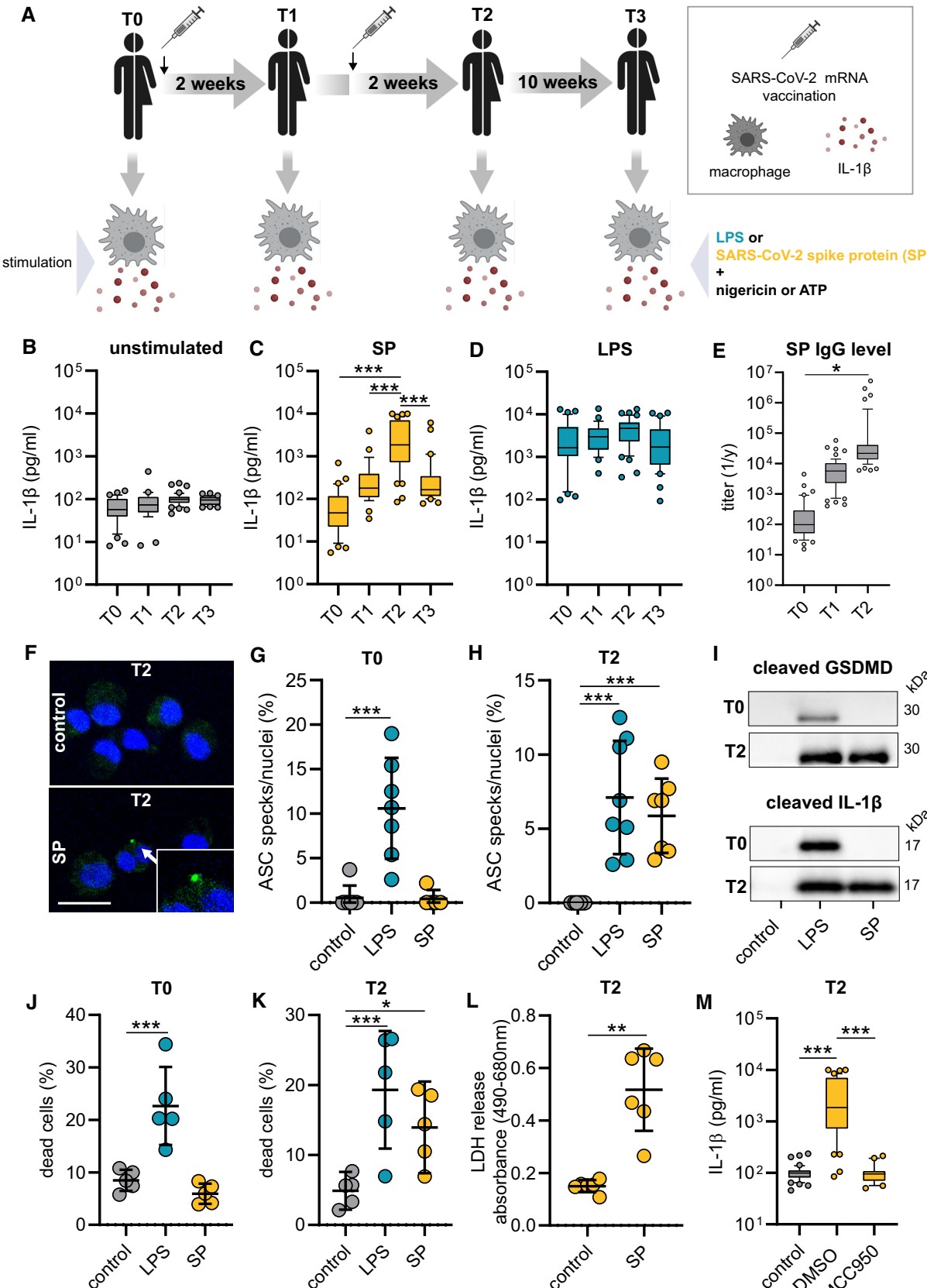

**Figure 1.**

**Table 1. Description of the ISAVak study cohort.**

| Number of individuals | T0 | 44 | Unvaccinated |
|---|---|---|---|
| | T1 | 43 | 2–4 weeks after 1st vaccination |
| | T2 | 54 | 2–4 weeks after 2nd vaccination |
| | T3 | 42 | 8–12 weeks after 2nd vaccination |
| | T4 | 12 | 24 h before 3rd vaccination |
| | T5 | 16 | 2–4 weeks after 3rd vaccination |
| Gender | Female | 57 | |
| | Male | 35 | |
| Vaccines used (Comirnaty® BNT162b2 / Spikevax® mRNA-1,273) | T0 | / | |
| | T1 | 2 / 41 | |
| | T2 | 7 / 47 | |
| | T3 | 7 / 35 | |
| | T4 | 0 / 7 | |
| | T5 | 9 / 2 | |

Further exploration of our RNA-seq data revealed C-type lectin domain family 4E (CLEC4E or Mincle) and CLEC4D as putative S-protein binding pattern recognition receptors (PRRs) to be upregulated in macrophages from vaccinated individuals (Fig 2B). CLRs have recently been identified as attachment receptors for SARS-CoV-2 on myeloid cells (Lu et al, 2021). To confirm surface expression of CLEC4D and CLEC4E, we quantified both receptors via flow cytometry. We were able to see increased expression levels of both CLR on post-vaccination macrophages compared to macrophages prior to SARS-CoV-2 vaccination (Fig 2C and D). To confirm CLR-dependent signaling of the S-protein, we performed blocking experiments by preincubating affinity purified CLR ectodomains [CLEC4E, CLEC4D or mannose binding lectin (MBL)] with the S-protein followed by stimulation of T2 macrophages and subsequent quantification of IL-1β. Incubation of the S-protein with the three CLRs inhibited IL-1β secretion indicating that these receptors are involved in S-protein dependent signaling (Figs 2E and F, and EV1E). Preincubation of CLRs with LPS, a Toll-like receptor 4 (TLR4) agonist,

had no impact on LPS-induced inflammasome activation and IL-1β secretion (Fig 2F). Furthermore, we were able to show that the S-protein S1 subunit and not the S2 subunit is required to prime inflammasome activation in post-vaccination macrophages (Fig 2G and H). Collectively, these data confirm CLRs as mediators of S-protein-dependent signaling in macrophages of vaccinated individuals.

**SYK signaling initiates NLRP3 inflammasome activation**

Recognition of microbial ligands by myeloid expressed CLRs leads to intracellular signaling via CLR-coupled spleen tyrosine kinase (SYK), a member of the non-receptor tyrosine kinase family (Osorio & Reis e Sousa, 2011). A key function of SYK lies in the capability to activate nuclear factor κB (NF-κB) (Osorio & Reis e Sousa, 2011). SYK additionally can activate the NLRP3 inflammasome (Fig 3A) (Gross et al, 2009; Mocsai et al, 2010). We first examined SYK phosphorylation and were able to detect higher levels of tyrosine phosphorylation in macrophages from vaccinated individuals compared to those from unvaccinated individuals indicating that SARS-CoV-2 mRNA vaccination activates SYK-dependent signaling pathways (Figs 3B and C, and EV2A and B). SYK promotes upregulation of inflammasome components via NF-κB. Accordingly, we identified higher levels of NLRP3 and phosphorylated NF-κB (pNF-κB p65) in unstimulated T2 macrophages compared to T0 macrophages. Ex vivo stimulation with the S-protein led to further upregulation of NLRP3 in T2 macrophages only (Figs 3D and EV2C and D). LPS was used as a control stimulatory agent inducing similar levels of NLRP3 in both T0 and T2 macrophages. Treatment of T2 macrophages with KINK-1, a small molecule inhibitor of NF-κB, abrogated IL-1β secretion following S-protein stimulation, indicating that NF-κB signaling is potentially involved in SYK-mediated inflammasome formation (Fig 3E).

To definitively link SYK signaling to activation of the NLRP3 inflammasome, we treated S-protein exposed T2 macrophages with SYK inhibitors (entospletinib or R406) and quantified IL-1β in cell supernatants. Chemical inhibition of SYK blocked IL-1β secretion from T2 macrophages significantly (Fig 3F). Moreover, we observed a reduction of ASC speck formation in S-protein-stimulated primary macrophages upon R406 treatment (Fig 3G). Similar findings were made with S-protein-exposed phorbol-12-myristate-13-acetate (PMA)-activated SYKKO THP-1 macrophages generated with

**Figure 2. Transcriptional changes in macrophages of vaccinated individuals.**

A Biological processes of gene set enrichment analyses performed with differentially expressed genes (DEGs) identified by RNA-seq in macrophages from vaccinated (T2, n = 6) or unvaccinated (T0, n = 6) individuals.

B Volcano plot showing DEGs in the same samples. Negative log10 padj values are plotted against the log2 fold-change. Selected genes are labeled.

C, D CLEC4D (C) and CLEC4E (D) surface expression on patient-derived macrophages prior (n = 5 individuals) or post (n = 5 individuals) vaccination were quantified by flow cytometry. Graphs indicate the mean fluorescence intensity (MFI).

E, F SP or LPS as a control PAMP were incubated with affinity purified CLRs (ratio: 10 vs. 1 for SP) for 1 h and then added to primary T2 macrophages (n = 5 individuals). Subsequently, IL-1β was quantified in the supernatants. (E) shows absolute values for SP; (F) shows the fold change calculated with the corresponding unstimulated control for SP and LPS.

G Monocytes were isolated prior (n = 5 individuals) or after vaccination (n = 5 individuals) and differentiated to macrophages. IL-1β concentrations in supernatants of macrophages were measured after stimulation with the SP S1 subunit SP and nigericin.

H Monocytes were isolated prior (T0; n = 35 individuals) or after vaccination [2 weeks after 1st (T1; n = 28 individuals), 2 weeks (T2; n = 38 individuals) and 10 weeks (T3; n = 33 individuals) after 2nd vaccination] and differentiated to macrophages. IL-1β concentrations in supernatants of macrophages, stimulated with the SP S2 subunit SP are shown.

Data information: For statistical analysis, one- or two-way ANOVA with Tukey post-hoc test was used. Box plots indicate the median and the upper and lower quartile. Outliers are plotted as individual dots (outside the 10–90 percentile). Scatter dot plots show mean ± SD. *P < 0.05; **P < 0.01.

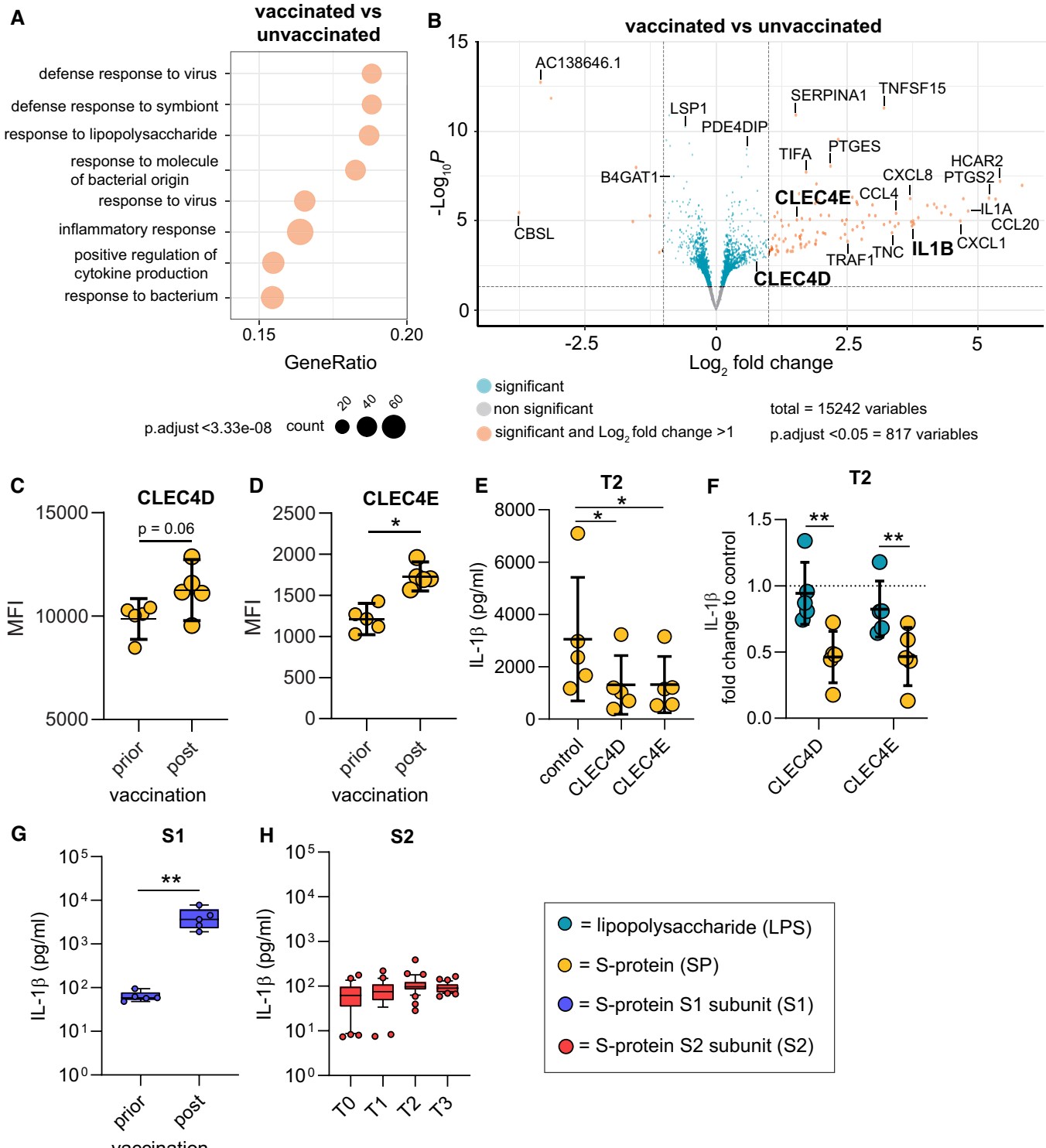

**Figure 2.**

CRISPR/Cas9 technology (Figs 3F and EV2E). These cells were also used to confirm SYK-dependent and S-protein-induced formation of ASC specks (Fig EV2F and G). In addition, cell death of stimulated T2 macrophages could be abrogated using a SYK inhibitor (Figs 3H and EV2H). Chemical SYK inhibition had only little effect on IL-1β

secretion of LPS-stimulated cells, which were used as a control (Fig EV2I).

SYK also drives the production of mitochondrial reactive oxygen species (mROS) in myeloid cells (Gross *et al*, 2009). Release of mROS has been linked to NLRP3 inflammasome formation (Zhou

et al, 2011). Consequently, we were able to show elevated levels of mitochondrial superoxide in S-protein-treated T2 macrophages but not in T0 macrophages using flow cytometry analyses of MitoSOX red-stained cells (Fig 3I and J). Macrophage treatment with the SYK inhibitor R406 inhibited superoxide release (Figs 3K and EV2J). Accumulation of mROS is coupled with changes in the mitochondrial membrane potential ($\Delta\Psi$m) and upregulation of superoxide dismutase 2 (SOD2) (Storz et al, 2005). Accordingly, S-protein stimulation led to SOD2 upregulation in T2 but not in T0 macrophages (Figs 3L and EV2K and Table EV1). As a result, we identified SYK-dependent $\Delta\Psi$m disruption in S-protein-treated T2 macrophages by quantifying intracellular tetramethylrhodamin-methylester (TMRM), a fluorescent dye that accumulates in mitochondria with intact membrane potential (Figs 3M and EV2L). Changes in $\Delta\Psi$m could be blocked by macrophage treatment with a SYK inhibitor (Figs 3N and EV2M). To confirm a link between mitochondrial damage and IL-1$\beta$ secretion, we treated cells with the mitochondrial ROS scavenger Mito-TEMPO, which led to a significant decrease of IL-1$\beta$ release (Fig EV2N).

To summarize these findings, SARS-CoV-2 mRNA vaccination leads to phosphorylation of SYK, which is required for mROS release, NLRP3 inflammasome activation, and pyroptosis after priming with the S-protein.

## SYK and NLRP3 mediate macrophage-dependent activation of T cells

As vaccines induce a specific adaptive cellular and humoral immune response required for protection against the target pathogen, we postulated that SYK-dependent inflammasome activation and pyroptosis in macrophages are associated with modulation of adaptive immunity. To address this hypothesis, we performed co-culture experiments of S-protein-stimulated and unstimulated T0 and T2 macrophages with autologous T cells to determine T cell differentiation and activation (Fig EV3A). We first focused on the quantification of effector memory T cells, which are the main responsible T cell subset required to target virus-infected cells (Kaech et al, 2002; Zanetti & Franchini, 2006). Interestingly, a 24-h co-culture of S-protein-stimulated T2 macrophages induced T cell differentiation from naïve to effector memory CD4$^+$ T cells, which was not observed for co-cultures with stimulated T0 macrophages (gating example in Fig EV3B) (Fig 4A and B). Macrophage pretreatment with the SYK inhibitor R406 or MCC950 reversed T cell maturation and PD-1 downregulation significantly (Figs 4C and D, and EV3C–F). Similar results could be observed with CD8$^+$ T cells (Figs 4E–H and EV3C–F). In line with these findings, we were able to detect significant downregulation of programmed cell death protein 1 (PD-1) in both CD4$^+$ and CD8$^+$ T cells co-cultured with stimulated T2 macrophages but not in those co-cultured with T0 macrophages (Fig 4I and J). PD-1 downregulation on T cells is known to enhance effector functions, cytokine secretion, and cell proliferation (Attanasio & Wherry, 2016). Moreover, we detected interferon-gamma (IFN-$\gamma$) and tumor-necrosis-factor-alpha (TNF-$\alpha$) as hallmark cytokines for CD4$^+$ and CD8$^+$ T cell activation at significantly higher levels in the supernatants of co-cultures containing S-protein-stimulated T2 macrophages compared to co-cultures performed with T0 macrophages (Figs 4K and EV3G). Cytokine secretion was abrogated by chemical inhibition of SYK or NLRP3 (Figs 4L and EV3H). To experimentally address the importance of IL-1$\beta$ on the activation of T cells, we repeated the experiments with the addition of a blocking antibody targeting IL-1$\beta$. The blockade of IL-1$\beta$ reversed the S-protein-induced effects on CD4$^+$ and CD8$^+$ naïve and effector memory T cells (Fig EV4A–D). In line, blocking IL-1$\beta$ led to higher expression levels of PD-1 on CD4$^+$ (Fig EV4E) and CD8$^+$ (Fig EV4F) T cells. To conclude, SYK, NLRP3 inflammasome activation, and IL-1$\beta$ secretion induced by SARS-CoV-2 mRNA vaccination are required for CD4$^+$ and CD8$^+$ T cell maturation ex vivo.

---

**Figure 3. Vaccination induces SYK phosphorylation required for NLRP3 inflammasome activation.**

A  SYK signaling scheme.

B  Phospho-SYK Immunofluorescence microscopy in macrophages before and after vaccination. Nuclei stained with DAPI. Scale bars indicate 20 μm.

C  Quantification of pSYK positive macrophages by flow cytometry before (n = 3 individuals) and 14 d after vaccination (n = 3 individuals). Total SYK was measured as control.

D  Western blot analysis of phospho-NF-κB and NLRP3 in cell lysates of macrophages (T0; T2) stimulated with LPS/N or SP/N.

E  Macrophages were incubated with IKK-β-inhibitor KINK-1 (n = 10 individuals) and SYK inhibitors entospletinib (n = 20 individuals), R406 (n = 13 individuals) prior to SP/nigericin stimulation. IL-1β concentrations measured with ELISA. DMSO-treated, SP/N-stimulated and untreated unstimulated macrophages were used as control (both n = 44 individuals).

F  IL-1β quantification in THP-1 wild-type cells incubated with SYK inhibitors entospletinib (n = 4 independent experiments) and R406 (n = 2 independent experiments) or in THP-1 SYKKO cells stimulated with SP/N (n = 5 independent experiments).

G  ASC formation quantified by immunofluorescence microscopy in SP/N stimulated primary macrophages after vaccination (n = 5 individuals).

H  Quantification of cell death in T2 macrophages (n = 4 individuals) treated with R406 prior to SP/N stimulation.

I–K  mROS quantified by flow cytometry using MitoSOX red staining at T0 and T2 in SP/N-stimulated macrophages without inhibitor (I [exemplary results], J [T0: n = 5 individuals; T2: n = 5 individuals]) and with R406 treatment (n = 4 individuals) (K).

L  Western blot analysis of SOD2 in macrophages at T0 and T2 treated with SP/N or LPS/N.

M  Mitochondrial membrane potential determined with flow cytometry of tetramethylrhodamine methyl ester (TMRM) stained macrophages at T0 (n = 5 individuals) and T2 (n = 5 individuals) stimulated with SP/N.

N  T2-macrophages (n = 4 individuals) were treated with the SYK-inhibitor R406.

Data information: For statistical analysis, one- or two-way ANOVA with Tukey post-hoc test was used. Box plots indicate the median and the upper and lower quartile. Outliers are plotted as individual dots (outside the 10–90 percentile). Scatter dot plots show mean ± SD. *P < 0.05; **P < 0.01, ***P < 0.001. (S-protein: SP, lipopolysaccharide: LPS, nigericin: N).

Source data are available online for this figure.

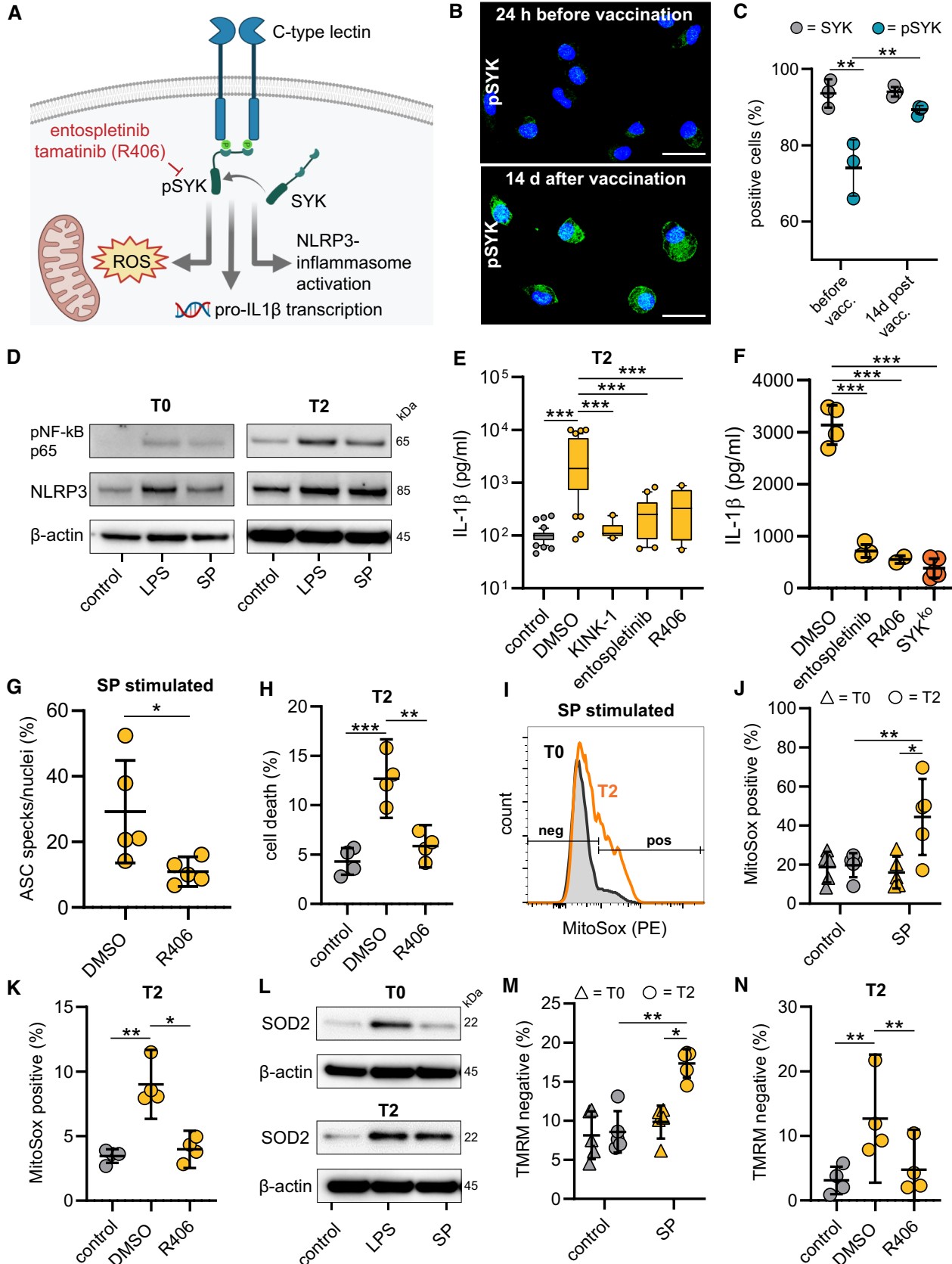

**Figure 3.**

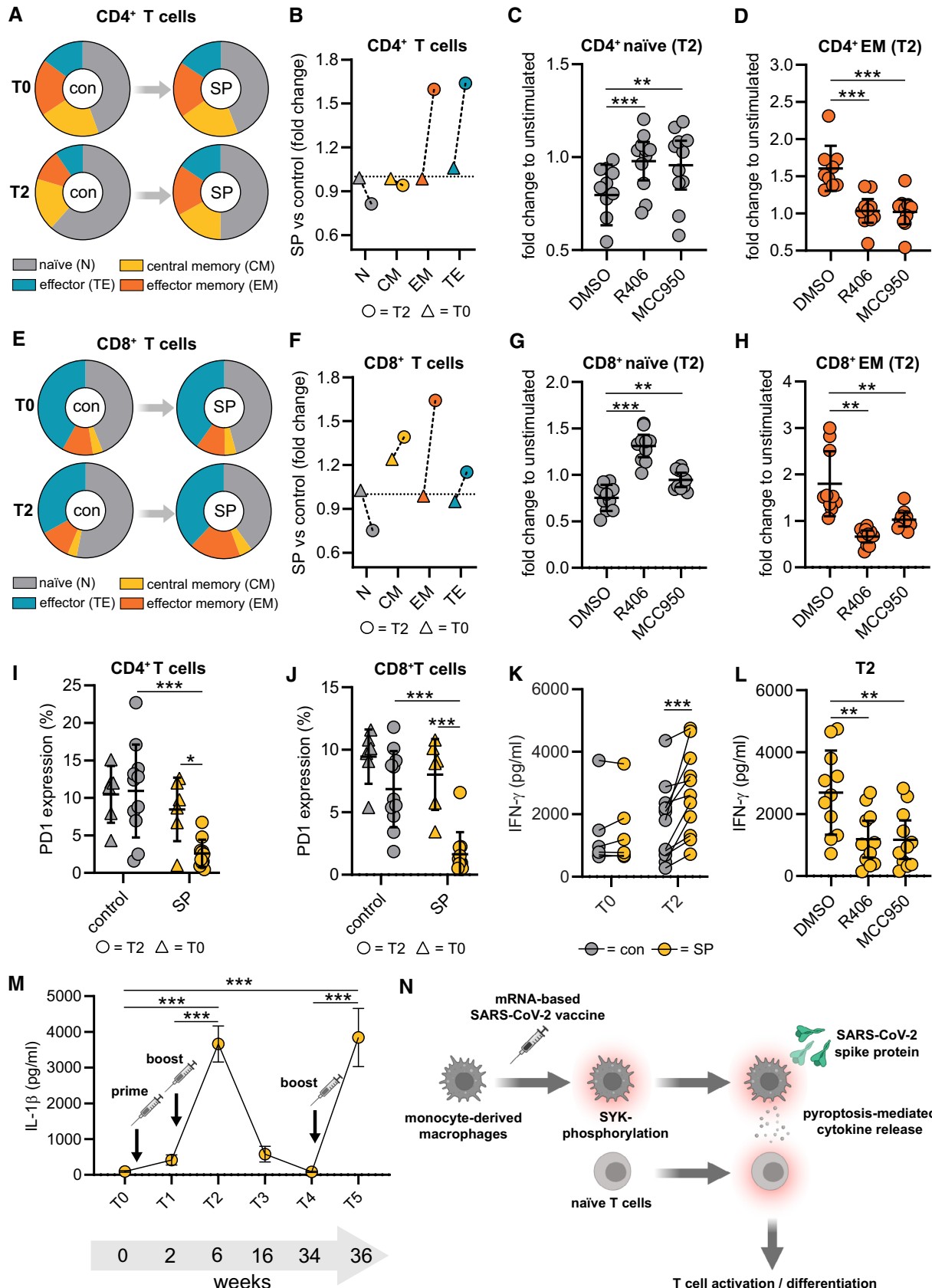

**Figure 4.**

**Figure 4. Macrophage SYK mediates activation of naïve T cells.**

A–H  (A) T cells at T0 ($n = 6$ individuals) and T2 ($n = 10$ individuals) were incubated with SP/N-stimulated macrophages. Cocultures of unstimulated macrophages were used as control (con). $CD4^+$ subpopulation analyses performed with flow cytometry (Experimental setup: Fig EV3). (B) Quantification of fold change of the $CD4^+$ T cell subpopulation upon SP/N stimulation (T0; T2). Dotted line indicates the baseline (fold change = 1.0). (C) Fold change of $CD4^+$ naïve T cell and $CD4^+$ effector memory T cell populations (D) upon co-culture of macrophages stimulated with SP/N in presence of MCC950 or R406. Identical analyses performed with $CD8^+$ T cells (E, F, G, H).

I, J  PD-1 expression was analyzed in $CD4^+$ (I) and $CD8^+$ T cells (J) by flow cytometry upon co-culture with SP/N-stimulated T0 ($n = 6$ individuals) or T2 ($n = 11$ individuals) macrophages. Co-cultures of unstimulated macrophages were used as control.

K, L  Interferon-γ quantified in supernatants of corresponding cocultures, untreated (T0 [$n = 6$ individuals] and T2 [$n = 11$ individuals]) (K) and treated with MCC950 or R406 (L) at T2 ($n = 11$ individuals).

M  IL-1β concentrations determined in SP/N-stimulated monocyte-derived macrophages at 34 weeks after 2nd vaccination (first booster) (T4 = 7 individuals) and 2 weeks after the 3rd vaccination (second booster) (T5 = 11 individuals).

N  Summary of main findings.

Data information: Statistical analysis: one or two-way ANOVA with Tukey *post hoc* test was used. Scatter dot plots show mean ± SD. SEM is shown for (M). *$P < 0.05$; **$P < 0.01$, ***$P < 0.001$. (S-protein: SP, lipopolysaccharide: LPS, nigericin: N).

## IL-1β levels detected after second booster vaccination

Having identified NLRP3-dependent secretion of IL-1β as an important trigger for T cell activation in SARS-CoV-2 mRNA vaccinated individuals, we were interested in macrophage derived IL-1β levels after the third dose (second booster) vaccination applied in our cohort six months after the first booster (Fig 4M and Table 1). We first tested for IL-1β secretion shortly prior to the second booster (T4) and found no or very little IL-1β secretion in S-protein-simulated macrophages. Levels were comparable to those detected in unvaccinated T0 macrophages (Fig 4M). However, in macrophages isolated 14 days after the second booster vaccination (T5), IL-1β levels were significantly higher than in macrophages after the first vaccination at T1 and mounted up to concentrations detected in T2 macrophage supernatants (Fig 4M). This indicates that peripheral blood monocytes remain primed for several months after first and second vaccination allowing for robust inflammasome activation and strong IL-1β secretion once a single second booster vaccination is applied.

# Discussion

In this work, we discovered a CLR-, SYK-, and NLRP3-dependent signaling pathway involved in macrophage activation following SARS-CoV-2 mRNA vaccination. There is growing evidence that cell surface expressed CLRs function as myeloid cell-interacting partners for SARS-CoV-2 promoting pro-inflammatory responses in COVID-19 (Lempp *et al*, 2021; Lu *et al*, 2021). SYK coupled to CLRs has been known as a key component of innate recognition primarily for fungal pathogens and mycobacteria (Gross *et al*, 2009; Mocsai *et al*, 2010). In addition, the CLR CLEC5A binds flaviviruses for release of pro-inflammatory cytokines (Chen *et al*, 2008). Regarding vaccines and vaccine adjuvants, which traditionally engage Toll-like receptor-dependent innate immune signaling, CLRs and SYK have been proposed as attractive alternative molecular targets for activation of innate immune cells (Pulendran *et al*, 2021). However, proof of principle had not been shown yet for vaccines applied in humans. Our data emphasize utility of this signaling pathway for mRNA vaccines, which demonstrate excellent safety profiles.

Stimulation of gene expression programs that lead to the production of pro-inflammatory cytokines is key to direct a particular and long-lasting adaptive immune response. In both vaccines and natural infections, inflammasome-derived IL-1β and associated receptors play a key role in transmitting stimulatory signals between innate and adaptive immune cells (Ichinohe *et al*, 2009; Reed *et al*, 2013; Munoz-Wolf & Lavelle, 2018; Van Den Eeckhout *et al*, 2020; Pulendran *et al*, 2021). Mice deficient for ASC, caspase-1, or IL-1 receptors fail to raise effector specific $CD4^+$ T cells and neutralizing antibodies against viral pathogens (Van Den Eeckhout *et al*, 2020). However, due to its highly inflammatory nature, IL-1β is produced and activated under strict regulation of inflammasomes (Broz & Dixit, 2016). Despite strong transcriptional upregulation of IL-1β in circulating monocytes of vaccinated individuals, two additional stimulatory signals were required for NLRP3 inflammasome activation, induction of necrotic cell death and secretion of mature IL-1β. A mechanism that may guarantee local instead of systemic inflammation at the expression site of mRNA-encoded S-protein. We also found S-protein-dependent maturation of IL-1β to be selective with no or very little IL-1β secreted in stimulated macrophages of non-vaccinated individuals, which stood in stark contrast to findings made in LPS/TLR4-stimulated cells. It seems that vaccine-dependent pre-activation of monocytes is not required for LPS-driven inflammasome processing *ex vivo*. One reason for this may be the chronic exposure of the human innate immune system to LPS via gut microbiota, which may lead to non-selective inflammasome activation in macrophages of both vaccinated and unvaccinated individuals.

Selective stimulation of primed immune cell subsets is a highly favorable feature of vaccines and vaccine adjuvants (Pulendran *et al*, 2021). Our data show that this can be achieved by exploiting signaling via CLR-coupled SYK, which represents a novel regulatory hub capable of differentiating between primed and unprimed macrophages. Furthermore, we observed $CD4^+$ and $CD8^+$ T cell stimulatory capacity exclusively in post-vaccination macrophages showing that SYK signaling also translates into selective stimulation of adaptive immunity, which is crucial for durable protection from severe COVID-19.

It is important to note that collaborative signaling between CLRs and TLRs has been described for some PAMPs, and this mechanism may further increase selective signaling of specific ligands in mRNA vaccination (Osorio & Reis e Sousa, 2011). We could recently show that TLR2 is upregulated in COVID-19 patient-derived macrophages, which promotes inflammasome activation in these cells when stimulating *ex vivo* (Theobald *et al*, 2021). In our transcriptome study of

monocytes isolated following vaccination, we did not detect TLR2 upregulation. Whether CLR- and TLR-2-dependent signaling is interconnected and involved in SYK-mediated inflammasome activation in macrophages requires further investigation. Additional integration of TLR signaling, for example, via RNA-dependent stimulation of TLR7 may also enhance local inflammation in mRNA vaccines (Teijaro & Farber, 2021).

There is growing evidence that third and fourth dose booster vaccinations potentiate and broaden adaptive immune responses towards wild-type SARS-CoV-2 and variants of concern (Munro et al, 2021). Innate immunity may play an important role in this process, which requires sufficient stimulation of T cell subsets by activated myeloid cells. Our data show potent secretion of the pro-inflammatory cytokine IL-1β from macrophages isolated two weeks after the third dose mRNA booster vaccination. IL-1β levels largely exceeded values detected after the first SARS-CoV-2 mRNA vaccination. Thus, innate immune cells seem to remain activated for several months after the first two vaccinations. Our mechanistic data can now be further exploited to decipher the potential of SYK/NLRP3 dependent signaling in future prime/boost vaccination strategies.

To conclude, we have identified NLRP3 inflammasome activation and pyroptosis in macrophages of SARS-CoV-2 mRNA vaccinated individuals as a novel innate immune trigger shaping secondary effector and memory adaptive responses (Fig 4N). Vaccination induces SYK and NF-κB phosphorylation in circulating monocyte-derived macrophages, which undergo a highly pro-inflammatory type of cell death once exposed to the SARS-CoV-2 S-protein. Considering the short lifespan of blood-derived monocytes, the observed effects are long-lived and can be enhanced with repetitive antigen exposure.

# Materials and Methods

### Blood samples and CD14$^+$ monocyte isolation

Blood samples were obtained at different time-points from healthy, unvaccinated donors or donors who had received one, two or three doses of SARS-CoV-2 mRNA vaccine, respectively (Table 1). COVID-19 vaccines were Comirnaty (Pfizer, BioNTech SE) or/and, Spikevax (Moderna, Inc., Cambridge, MA, USA). For all human samples, written informed consent was obtained in accordance with the WMA declaration of Helsinki and the experiments conformed to the principles set out in the Department of Health and Human Services Belmont Report. The study was approved by the ethics committee of Cologne (Reference number 21-1283). Only adults were included in the study. Individuals with COVID-19 infections after vaccination were excluded.

Purification of PBMCs (peripheral blood mononuclear cells) and isolation of CD14$^+$ cells were performed as described previously (Theobald et al, 2021). The desired number of CD14$^+$ cells was seeded into the corresponding culture vessels and cultured for 5 days in Roswell Park Memorial Institute (RPMI) 1640 Medium (Thermo Fisher Scientific, Waltham, MA, USA) containing 10% fetal bovine serum (FBS) (Thermo Fisher) and 50 ng/ml M-CSF (Miltenyi Biotec, Bergisch Gladbach, Germany) for macrophage differentiation at 37°C and 5% $CO_2$.

### Generation of THP-1 Syk knockout cells via CRISPR/Cas9

To generate a Cas9 nuclease expressing THP-1 cell line, $1 \times 10^6$ THP-1 cells (CVCL_0006) were transduced twice with third-generation lentiviral Cas9-eGFP particles (MOI 5, Thermo Fisher) via spin infection at 800 rpm for 120 min at 37°C. Afterwards, eGFP high-expressing single cell clones were selected by eGFP-mediated fluorescence activated cell sorting. Lentiviral particles containing four separate guide RNAs targeting the SYK gene were acquired from Thermo Fisher. In a 96-well plate, $1 \times 10^4$ THP-1 Cas9-eGFP cells were transduced with the lentiviral particles (MOI 10). Spin infection was performed at 800 rpm for 90 min at 37°C. 72 h p.i., cells were washed 2 × with PBS and positive transduced cells were selected via addition of 0.75 μg/ml puromycin (Carl Roth, Karlsruhe, Germany) into RPMI media for at least 7 days. The spent medium was replaced with fresh medium containing puromycin after 3 days. All cells were authenticated and tested for mycoplasma contamination.

### Cloning and expression of the spike protein

The plasmid coding for the SARS-CoV-2 spike protein was a gift from Jason S. McLellan (Hsieh et al, 2020). To produce a stable soluble trimeric S-protein, the ectodomain was amplified by PCR (MN908947; AA: 1–1,208; RRAR to GSAS; F817P; A892P; A899P; A942P; K986P; V987P;139 kDa). The same was performed for the soluble C-type lectin receptors: CLEC4E/MINCLE (NP_055173) AA: 41–219 N-terminal signal peptide followed by a Twin-Strep-tag; CLEC4D/Dectin-3 (NP_525126) AA: 43–215 N-terminal signal peptide followed by a Twin-Strep-tag; CLEC5A/MDL-1 (NP_037384) AA: 26–188 N-terminal signal peptide followed by a Twin-Strep-tag; SFTPD/SP-D (NP_003010) AA: 259–375 N-terminal signal peptide followed by a Twin-Strep-tag; Mannan-binding lectin MBL (AF360991) AA: 126–.0.248 N-terminal signal peptide followed by a Twin-Strep-tag. The PCR product was digested with the appropriate restriction enzymes and cloned into a modified sleeping beauty transposon expression vector containing for the ectodomain a C-terminal T4 fibritin (foldon) followed by a Twin-Strep-tag. For recombinant protein production, a stable HEK293 EBNA cell line was generated applying the sleeping beauty transposon system (Kowarz et al, 2015). Briefly, the expression construct was transfected into HEK293 EBNA cells and after selection with puromycin the cells were expanded in triple flasks and protein production was induced by addition of doxycycline. Cell supernatants were harvested every 3 days, filtered, and the recombinant proteins purified via Strep-Tactin®XT (IBA Lifescience, Göttingen, Germany) resin. Proteins were eluted by biotin containing TBS-buffer (IBA Lifescience), dialyzed against TBS-buffer, analyzed by SDS-PAGE, and aliquots stored at either 4°C or −80°C. All reusable materials were treated with 1 M NaOH for 2 h to remove any LPS contamination.

### SARS-CoV-2 spike IgG binding ELISA

To analyze the interaction of IgG with SARS-CoV-2 spike protein in human plasma, high binding 96 well ELISA plates (Corning Inc., Corning, NY, USA) were coated with SARS CoV-2 spike protein (2 μg/ml) in Dulbecco's Phosphate Buffered Saline (DPBS) (Thermo Fisher) at 4°C overnight. Next, the plates were washed 3 × with

DPBS and blocked with DPBS containing 10% FBS for 60 min at RT. The plates were washed 4 × and plasma was applied in 5-fold dilutions for 60 min at RT. Thereafter, the plates were washed 4 × and incubated with horseradish peroxidase-conjugated goat anti-human IgG antibody (Jackson ImmunoResearch, West Grove, PA, USA) (1:500) in DPBS for 60 min at RT. The plates were washed 4 × and bound horseradish peroxidase was detected using 1-StepTM Ultra TMB-ELISA solution (Thermo Fisher) for 10–15 min at RT. The reaction was stopped with 2N sulfuric acid (Carl Roth) and absorbance was measured (OD 450–570 nm) in a microplate reader (Hidex Oy, Turku, Finland).

### *Ex vivo* stimulation of macrophages

Culture medium of differentiated macrophages was exchanged after 5 days of incubation. For inhibitor experiments, cells were preincubated with either 0.1% DMSO (Merck, KGaA, Darmstadt, Germany), MCC950 (10 μM) (Merck), KINK-1 (5 μM) (Merck), MitoTEMPO (10 μM) (Merck), entospletinib (5 μM) (Gilead Sciences, Inc., Foster City, CA, USA), or tamatinib/R406 (5 μM) (InvivoGgen, San Diego, CA, USA) in RPMI medium for 2 h at 37°C and 5% $CO_2$. Thereafter or without inhibitors, cells were stimulated with lipopolysaccharide (LPS) (5 μg/ml) (Merck) or SARS-CoV-2 spike protein (0.1 μg/ml) and incubated for 4 h at 37°C and 5% $CO_2$ to prime the inflammasome process. Inflammasome activation and IL-1β secretion were initiated by addition of nigericin (5 μM) (Merck) or ATP (5 mM) (Thermo Fisher) and the cells were further incubated for 2 h (Theobald *et al*, 2021).

### Stimulation of THP-1 macrophages

THP-1 cell, a human monocytic cell line, was cultured in RPMI medium (Thermo Fisher) containing 10% FBS (PAN Biotech). For differentiation into macrophages cells were incubated with 20 nmol Phorbol 12-myristate 13-acetate (PMA) (Merck) for 24 h. After 24 h, medium was exchanged, and cells were cultured for another 24 h without PMA. All cells were cultured at 37°C with 5% $CO_2$. Stimulation of differentiated THP-1 macrophages was performed as described before (Eisfeld *et al*, 2021).

### Quantification of IL-1β

Quantitative detection of IL-1β was achieved using the IL-1 beta Human Uncoated ELISA Kit (Thermo Fisher) and performed according to the manufacturer's instructions. Supernatants of primary macrophages and THP-1 cells were diluted in ELISA diluent 1:75 and 1:50, respectively. All samples were measured in technical duplicates and absorbance was measured in a microplate reader (Hidex Oy). All data were analyzed with Microsoft Excel (Microsoft, Redmond, WA, USA) and GraphPad Prism 8.0.2 (GraphPad, San Diego, CA, USA).

### Immunoblot analysis

Human macrophages were isolated as described before and seeded on 24-well plates with $2.5 × 10^6$ cells per well. Stimulation was performed as described before (Eisfeld *et al*, 2021). Cell supernatant was precipitated with chloroform and methanol as described before

(Eisfeld *et al*, 2021). Adherent cells were lysed with RIPA buffer (Thermo Fisher) and protein concentration was measured with the PierceTM BCA Protein Assay Kit (Thermo Fisher). For Western blot analysis with supernatant, experiments were normalized to the number of input cells added to the respective wells in the experiment. Proteins or precipitated supernatant were separated via SDS-PAGE (NuPAGE 4–12%, Bis-Tris gel,12 well, Thermo Fisher) and subsequently blotted on PVDF membranes (TransBlot Turbo Transfer System) (Bio-Rad Laboratories, Inc., Hercules, CA, USA). The membranes were blocked in 5% Milk/TBST or BSA/TBST for 1 h and incubated with a primary antibody over night at 4°C. The next day, membranes were washed and incubated with the respective secondary antibody for 1 h and subsequently visualized by chemoluminescence-based detection. The following antibodies were used: anti-p-NF-kB p65 (93H1; 1:1000), anti-NLRP3 (D4D8T; 1:1000), anti-β-actin (13E5; 1:5000), anti-SOD2 (E-10; 1:1000), anti-cleaved GSDMD (E7H9G; 1:000), anti-cleaved IL-1β (D3A3Z; 1:1000), HRP-linked anti-mouse IgG (7,076; 1:5000), HRP-linked anti-rabbit IgG (7,074; 1:5000) (Santa Cruz Biotechnology, Dallas, TX, USA and Cell Signaling Technology, Danvers, MA, USA). All data were analyzed with ImageJ.

$SYK^{KO}$ and scramble (scr) control THP-1 monocytes were lysed in RIPA buffer (Cell Signaling) containing protease inhibitor (Complete mini EDTA-free; Roche) and phosphatase inhibitor cocktail (PhosStop; Roche). Lysates were clarified by centrifugation at 13,000 rpm for 20 min at 4°C. A total of 30 μg protein per sample was separated via SDS-PAGE and transferred to a nitrocellulose membrane (Amersham Biosciences). The membrane was cut below 50 kDa, according to the protein standard (Precision Plus Protein Dual Color Standard; Bio-Rad). The upper part was incubated with the primary antibody to SYK (D3Z1E; Cell Signaling) and the lower part with the antibody against GAPDH (D16H11; Cell Signaling) overnight at 4°C. The following day, blots were rinsed and incubated with secondary antibody IRDye® 680LT anti-Rabbit IgG (LI-COR) for 1 h at RT. Fluorescence signal was detected via LI-COR ODYSSEY CLx fluorescence imaging system (LI-COR).

### ASC-SPECK immunofluorescence staining

Human macrophages were isolated as described before. Human macrophages or THP-1 cells were seeded on a pre-coated (1 mg/ml fibronectin, Merck) 8-well chamber slide (Corning Inc.) with $8 × 10^4$ cells per well. Stimulation was performed as described before. Cells were washed 2 × with DPBS and fixated with DPBS containing 4% PFA for 15 min at RT. Cells were washed 2 × and permeabilized with DPBS containing 5% FBS, 0.1% Tween-20 (Carl Roth) and 0.1% Triton X-100 (Merck) for 1 h. Anti-ASC Alexa488 antibody (B-3; 1:1000) (Santa Cruz Biotechnology) was added overnight at 4°C in DPBS containing 3% FBS, 0.1% Tween-20 and 0.1% Triton X-100. Next, cells were washed 2 × and incubated with DAPI (1:1000) in DPBS for 10 min at room temperature. Cells were washed 2 × and the slides were dried at RT. Afterwards, mounting medium (Thermo Fisher) and cover slips were added to the slides. Images were acquired on an Olympus Fluoview FV1000 (Olympus K.K., Tokio, Japan) or Leica TCS SP8 (Leica Biosystems, Nussloch, Germany) confocal microscope with 60x objective using the same microscope settings for all pictures. All data were analyzed with ImageJ and GraphPad Prism 8.0.2 (GraphPad).

## (Phospho-)SYK immunofluorescence staining

Human macrophages were isolated as described before and seeded on a fibronectin precoated 8-well chamber slide (Corning Inc.) with $8 \times 10^4$ cells per well. Right before fixation, cells were incubated with RIPA buffer (Thermo Fisher) to stop phosphatase activity. Next, cells were washed 2x with DPBS and fixed with DPBS containing 4% PFA for 15 min at RT. Cells were washed 2 × and permeabilized with DPBS containing 5% FBS, 0.1% Tween-20 (Carl Roth) and 0.1% Triton X-100 (Merck) for 1 h. Anti-pSYK antibody (EP573-4; 1:100) or anti-SYK antibody (EP573Y; 1:100) (Abcam, Cambridge, UK) were added overnight at 4°C in DPBS containing 3% FBS, 0.1% Tween-20 and 0.1% Triton X 100, respectively. The next day, cells were washed 2 × and incubated with anti-rabbit-Alexa488 antibody (ab150077; 1:200) (Abcam) in DPBS containing 3% FBS, 0.1% Tween-20 and 0.1% Triton-X 100 for 2 h at RT. Next, cells were washed 2 × and incubated with DAPI (1:1000) in DPBS for 10 min at room temperature. Cells were washed 2 × and the slides were dried at RT. Afterward, mounting medium (Thermo Fisher) and cover slips were added to the slides. Pictures were acquired on a Leica TCS SP8 confocal microscope (Leica Biosystems) with 60x objective using the same microscope settings for all pictures. All data were analyzed with ImageJ and GraphPad Prism 8.0.2 (GraphPad).

## (Phospho-)SYK flow cytometry analysis

Human macrophages were isolated as described before and seeded on fibronectin precoated 24-well plates with $2 \times 10^5$ cells per well. Before fixation, cells were incubated with RIPA buffer (Thermo Fisher) to stop phosphatase activity. Next, cells were stained with eBioscience™ Intracellular Fixation & Permeabilization Buffer Set (Thermo Fisher) according to the manufacturer's instructions. Anti-pSYK or anti-SYK antibody (1:100) (Abcam) was added overnight at 4°C and anti-rabbit-Alexa488 antibody (1:200) (Abcam) was added for 2 h at RT. Cells were analyzed with an BD Bioscience Canto 2 cytometer (BD, Heidelberg, Germany). Cytometer settings were kept the same for all experiments and data was analyzed with FlowJo software (BD).

## Transcriptome sequencing (RNA-Seq) and bioinformatics analysis

Human macrophages were isolated as described before and seeded on fibronectin precoated 24-well plates with $2.5 \times 10^5$ cells per well. Stimulation was performed as described before. Cells were washed 2 × with DPBS and isolation of the RNA was performed with the mirVana miRNA isolation kit (Thermo Fisher) according to the manufacturer's instructions. RNA-Seq library prep was performed with 100 ng total RNA input and the NEBNext Ultra RNA library prep protocol (New England Biolabs, Ipswich, MA, USA) according to standard procedures. Libraries were validated and quantified (Tape Station 4,200, Agilent Technologies, Santa Clara, CA, USA).

All libraries were quantified by using the KAPA Library Quantification Kit (VWR International, Radnor, PA, USA) and the 7900HT Sequence Detection System (Applied Biosystems, Foster City, CA, USA). Sequencing was done with NovaSeq6000 sequencers (Illumina, San Diego, CA, USA) with a PE100bp read length aiming at 50 M reads/sample (RNA-Seq) or a SR50bp read length aiming at 5 M reads/sample (small RNA). Demultiplexing and FastQ file generation was performed using Illumina's bcl2fastq2 software (v2.20.0).

RNA-seq was performed with a directional protocol. Quality control, trimming, and alignment were performed using the nf-core (Ewels et al, 2020) RNA-seq pipeline (v3.0). Details of the software and dependencies for this pipeline can be found at the referenced DOI for the pipeline and: https://github.com/nf-core/rnaseq/blob/master/CITATIONS.md. The reference genome sequence and transcript annotation used were Homo sapiens genome GRCh38 from Ensembl version 103. Differential expression analysis was performed in R version 4.1.1 (2021-08-10) (R Core Team, 2021) with DESeq2 v1.32.0 to make pairwise comparisons between groups. Log Fold Change shrinkage estimation was performed with ashr (Stephens, 2017). Only genes with a minimum coverage of 10 reads in 6 or more samples from each pairwise comparison were considered as candidates to be differentially expressed. Genes were considered deferentially expressed if they showed a log2(Fold Change) > 1 and were below an FDR of 0.05. Genes with a minimum coverage of 10 reads in 6 or more samples from each pairwise comparison were included in functional enrichment analyses and considered as the "gene universe" for overrepresentation-based analyses. Functional enrichment analysis was performed with clusterProfiler v4.0.5 (Yu et al, 2012). Gene Set Enrichment Analysis (GSEA) (Subramanian et al, 2005) was performed using the fgsea v1.18.0 algorithm. The P-value cutoff was 0.05 by permutation (using the default values in clusterProfiler), for genes rank ordered by log2 (Fold Change).

The GSEA dotplot was plotted with enrichplot v1.12.2. The Volcano plot was plotted with a modified version of the EnhancedVolcano function from EnhancedVolcano v1.13.2.

## CLR inhibition assay

For the CLR inhibition assay, affinity purified CLEC proteins were preincubated with spike protein at a ratio of 10:1 at 37°C for 1 h. Afterwards, the mixture was added to the T2 macrophages and treated as already described.

## LDH cytotoxicity assay

Human macrophages were isolated as described before and seeded on 96-well plates with $1 \times 10^5$ cells per well. Stimulation was performed as described before (Eisfeld et al, 2021). All samples were measured in technical triplicates after 24 h incubation with nigericin (5 μM). Quantitative analysis of lactate dehydrogenase (LDH) as a reliable indicator of cytotoxicity was performed with the CyQUANT™ LDH Cytotoxicity Assay Kit (Thermo Fisher) according to the manufacturer's instructions.

## Quantification of mitochondrial ROS

Human macrophages were isolated as described before and seeded on fibronectin precoated 24-well plates with $2 \times 10^5$ cells per well. Stimulation was performed as described before. To analyze mitochondrial ROS production, the cells were detached from the wells, washed 2 × with DPBS containing 5% FBS and 5 mM EDTA, and stained with MitoSOX™ Red mitochondrial superoxide indicator (5 μM, Thermo Fisher) for 15 min at 37°C and 5% $CO_2$. Cells were

washed 2 × and analyzed with an BD Bioscience Canto 2 cytometer. Cytometer settings were kept the same for all experiments and data was analyzed with FlowJo software (BD).

## TMRM staining

Human macrophages were isolated, seeded on precoated 24-well plates with $2 \times 10^5$ cells per well, and stimulated as described before. To analyze the mitochondrial membrane potential, the cells were detached from the wells, washed 2 × with DPBS containing 5% FBS and 5 mM EDTA, and stained with TMRM (100 nM) (Thermo Fisher) for 30 min at 37°C and 5% $CO_2$. Cells were washed 2 × and cell viability was analyzed with an BD Bioscience Canto 2 cytometer. Cytometer settings were kept the same for all experiments and data were analyzed with FlowJo software (BD).

## Cell viability analysis

Human macrophages were isolated, seeded on precoated 24-well plates with $2 \times 10^5$ cells per well, and stimulated as described before. To analyze cell viability, the cells were detached from the wells, washed 2 × with DPBS containing 5% FBS and 5 mM EDTA, and stained with Zombie UV (1:500) (BioLegend, San Diego, CA, USA) for 15 min or 30 min at 37°C and 5% $CO_2$, respectively. Cells were washed 2 × and cell viability was analyzed with an BD Bioscience Canto 2 cytometer. Cytometer settings were kept the same for all experiments and data was analyzed with FlowJo software (BD).

## Co-culture of macrophages and T cells

For all experiments, autologous macrophages and T cells were used. On the day prior to macrophage isolation, total PBMCs were thawed, and T cells were activated for 24 h at 37°C using human IL-2 (1,000 U/ml, Miltenyi Biotec), in-house produced anti-CD3 (0.2 µg/ml) and anti-CD28 antibodies (0.1 µg/ml, both kind gift from Dr. Markus Chmielewski, University of Cologne) in RPMI media supplemented with 10% FBS. 5 days prior to co-culture, macrophages were isolated and stimulated after 5 days of differentiation as mentioned before. After stimulation, macrophages and T cell cultures were washed 2 × with RPMI media supplemented with 10% FBS. Macrophage supernatant was stored at −80°C for cytokine detection. Co-cultures were set up in 1:1 ratio of macrophages to T cells in 96-well plate. For blocking IL-1β the specific antibody was added to the culture (1 µg/ml; Invivogen). Co-cultures were further incubated at 37°C for 24 h. T cells were collected for flow cytometry by pipetting and cell-free supernatant for cytokine detection was stored at −80°C.

## Flow cytometry

Cells were detached and afterwards blocked for 20 min at 4°C with DPBS containing 10% FBS and 5 mM EDTA. Cells were washed 1 × with DPBS containing 1% FBS and 5 mM EDTA. All antibodies were previously titrated for optimal working concentrations. Antibodies used in the study: Pacific Blue™ anti-human CD4 (317429); FITC anti-human CD45RA (304106); PE/Cyanine7 anti-human CD62L (304822); PerCP anti-human CD279 (329938); PE anti-human CD8a (301051) (1:100 dilution for all antibodies; Biolegend, San Diego,

**The paper explained**

**Problem**

Novel mRNA-based vaccines have had tremendous impact on the control of the COVID-19 pandemic. There is an increasing amount of knowledge on adaptive immune responses involving B and T cell activation following SARS-CoV-2 mRNA vaccination. However, sufficient induction of immune memory in these cells requires potent stimulation of the innate immune system and associated cell types such as macrophages.

**Results**

Here, we decipher a macrophage-dependent signaling pathway that involves activation of the spleen tyrosine kinase (SYK) and upregulation of C-type lectins, which can bind the SARS-CoV-2 S-protein. These events are prerequisites for cytokine release and inflammasome activation in macrophages, which in turn stimulate the formation of antigen-specific T cells. Macrophages remain in an activated state for several months allowing for rapid activation following booster vaccinations.

**Impact**

The elucidation of this signaling axis will help to tailor safe and efficient mRNA vaccines, conventional vaccines, and vaccine adjuvants for future applications.

CA, USA). PE anti-human CLEC4D (130–110-721; Miltenyi Biotech) and Alexa488 anti-human CLEC4E (sc-390,806 AF48, SantaCruz) were diluted 1:50. Cells were incubated with antibodies for 30 min at 4°C and washed afterwards once. Cytometer settings were kept the same for all experiments and data was analyzed with FlowJo software (BD).

## Quantification of IFNγ and TNFα

Quantitative detection of IFNγ and TNFα was achieved done using the IFNγ and TNFα human Uncoated ELISA Kit (Thermo Fisher) and performed according to the manufacturer's instructions. Supernatant of primary macrophages were diluted 1:10 in ELISA diluent. All samples were measured in technical duplicates and absorbance was measured in a microplate reader (Hidex). All data were analyzed with Microsoft Excel (Microsoft) and GraphPad Prism 8.0.2 (GraphPad).

## Statistical analysis

Statistical analysis was performed with GraphPad Prism 8.0.2 software (GraphPad). Statistical parameters (value of $n$, statistical calculation, etc.) are also provided in the figure legend. $P$-values less than or equal to 0.05 were considered statistically significant ($*P < 0.05$; $**P < 0.01$, $***P < 0.001$) For comparison of multiple groups, we used one or two way ANOVA depending on the comparison (for all groups homogeneity variance was tested) and the data-set with Tukey post-test. $T$-test with Welsh's corrections were used for comparison of two groups. Box plots indicate the median and the upper and lower quartile. Outliers are plotted as individual dots (outside the 10–90 percentile). Scatter dot plots show mean. Data points represent biological replicates. All experiments were conducted at least in technical duplicates. Samples

between the different groups were matched regarding age, sex and vaccination status. Apart from that, all samples were selected randomly, and all other information were blinded. Sample sizes were chosen depending on the expected variance and available samples.

## Data availability

The datasets produced in this study are available in the following databases: Transcriptome data: Gene Expression Omnibus, GSE200274 (https://www.ncbi.nlm.nih.gov/geo/query/acc.cgi?acc=GSE200274). All other data that support the findings of this study are available from the corresponding author on request.

Expanded View for this article is available online.

### Acknowledgements

We are grateful to vaccinated individuals for donating blood used in this investigation. We thank Prof. Manolis Pasparakis (University of Cologne) and Prof. Hamid Kashkar (University of Cologne) for helpful discussions during the preparation of the manuscript. We thank Dr. Markus Chmielewski (University of Cologne) for providing the anti-CD3/CD28 antibodies. We thank Jason Chhen and Edeltraud van Gumpel from the Translational Research Unit-Infectious Diseases (TRU-ID) for their technical help. Schemes were created with BioRender.com.

JR is supported by German Research Foundation (DFG) grant DFG RY 159/3-1; German Research Foundation (DFG) grant SFB1403; BMBF NaFoUniMed-Covid19 grant COVIM 01KX2021 and German Center for Infection Research (DZIF) grant TTU 02.806. AS is supported by the German Federal Ministry of Education and Research (BMBF) grant 01KI2108 and by the Cologne Clinician Scientist Program (CCSP), funded by the German Research Council (FI 773/15-1). AS and SJT are supported by the Koeln Fortune Program. AMH is supported by Else Kröner-Fresenius-Stiftung (EKFS) grant 2020_EKFK.19. MK is supported by German Research Foundation (DFG) grant FOR 2722. HW is supported by the Alexander von Humboldt Foundation. Open Access funding enabled and organized by Projekt DEAL.

### Author contributions

**Sebastian J Theobald:** Conceptualization; data curation; formal analysis; supervision; funding acquisition; validation; investigation; visualization; methodology; writing – original draft; project administration; writing – review and editing. **Alexander Simonis:** Conceptualization; data curation; formal analysis; supervision; funding acquisition; investigation; visualization; methodology; writing – original draft; project administration; writing – review and editing. **Julie M Mudler:** Conceptualization; data curation; formal analysis; investigation; visualization; writing – original draft; writing – review and editing. **Ulrike Göbel:** Conceptualization; data curation; software; formal analysis; investigation; visualization; methodology; writing – review and editing. **Richard Acton:** Data curation; software; formal analysis; validation; investigation; visualization; methodology; writing – review and editing. **Viktoria Kohlhas:** Resources; data curation; investigation; methodology. **Marie-Christine Albert:** Resources; data curation; investigation. **Anna-Maria Hellmann:** Resources; investigation. **Jakob J Malin:** Resources; investigation. **Sandra Winter:** Investigation. **Michael Hallek:** Resources. **Henning Walczak:** Resources; investigation. **Phuong-Hien Nguyen:** Resources; investigation. **Manuel Koch:** Resources; funding acquisition; validation; investigation; methodology; writing – review and editing. **Jan Rybniker:** Conceptualization; resources; supervision; funding acquisition; validation; investigation; methodology; writing – original draft; project administration; writing – review and editing.

In addition to the CRediT author contributions listed above, the contributions in detail are:

SJT, AS, and JMM contributed samples, planned and performed experiments, analyzed data, and wrote the manuscript; UG and RA performed the sequencing analysis and integration; VK, MCA, AMH, JJM, and SW performed experiments and assisted with the selection of vaccinated individuals; MH, HW, and PHN provided resources and discussed data, MK, JR planned experiments, analyzed and discussed data; JR directed the study and wrote the manuscript.

### Disclosure and competing interests statement
The authors declare that they have no conflict of interest.

### For more information

- https://www.who.int/emergencies/diseases/novel-coronavirus-2019
- https://www.who.int/publications/m/item/draft-landscape-of-covid-19-candidate-vaccines
- https://www.fda.gov/emergency-preparedness-and-response/coronavirus-disease-2019-covid-19/covid-19-vaccines
- https://www.ema.europa.eu/en/human-regulatory/overview/public-health-threats/coronavirus-disease-covid-19/covid-19-latest-updates
- https://innere1.uk-koeln.de/forschung/arbeitsgruppen-labore/translational-research-unit-infectious-diseases/translational-research-unit-infectious-diseases-en/

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
