## [Review Process File · EMBO Molecular Medicine]

Spleen tyrosine kinase mediates innate and adaptive immune crosstalk in SARS-CoV-2 mRNA vaccination

Sebastian Theobald, Alexander Simonis, Julie Mudler, Ulrike Göbel, Richard Acton, Viktoria Kohlhas, Marie-Christine Albert, Anna-Maria Hellmann, Jakob Malin, Sandra Winter, Michael Hallek, Henning Walczak, Phuong-Hien Nguyen, Manuel Koch, and Jan Rybniker

DOI: [10.15252/emmm.202215888](https://doi.org/10.15252/emmm.202215888)

Corresponding author: Jan Rybniker (jan.rybniker@uk-koeln.de)

Review Timeline:

Submission Date:	16th Feb 22
Editorial Decision:	24th Mar 22
Revision Received:	18th May 22
Editorial Decision:	8th Jun 22
Revision Received:	11th Jun 22
Accepted:	14th Jun 22

Editor: Zeljko Durdevic

Transaction Report:

24th Mar 2022

Dear Dr. Rybniker,

Thank you for the submission of your manuscript to EMBO Molecular Medicine, and please accept my apologies for the delay in getting back to you. We have now received feedback from two of the three reviewers who agreed to evaluate your manuscript. Given that referee #3 will unfortunately not be able to return his/her report in a timely manner, and that both referees #1 and #2 are overall positive, we prefer to make a decision now in order to avoid further delay in the process. Should referee #3 provide a report, we will send it to you, with the understanding that we will not ask you extensive experiments in addition to the ones required in the enclosed reports from referee #1 and #2. As you will see from the reports below, the referees acknowledge the interest of the study but also raise important concerns that should be addressed in a major revision.

We would welcome the submission of a revised version within three months for further consideration. Please let us know if you require longer to complete the revision.

Please use this link to login to the manuscript system and submit your revision: <https://embomolmed.msubmit.net/cgi-bin/main.plex>

I look forward to receiving your revised manuscript.

Yours sincerely,

Zeljko Durdevic

***** Reviewer's comments *****

Referee #1 (Comments on Novelty/Model System for Author):

The work was performed to a high quality standard for most of the experiments, including appropriate controls. Some of the findings reported (S protein being recognized by CLR receptors, SYK signalling mediating NLRP3 activation -though not shown in the context of SARS-CoV-2-) are published. However, the novelty arises from the use of relevant human samples with different vaccination status, indicating that vaccine primes only S protein-dependent NLRP3 activation. The work is of medical interest. The statistical tests are appropriate.

Referee #1 (Remarks for Author):

Theobald and colleagues investigate the effect of vaccination on inflammasome activation, and the effect on macrophage T- cell activation. Authors report that only cells from vaccinated individuals response to S-protein challenge releasing iL1B, and resulting in pyrptosis. RNA-seq data showed that the S-protein receptors and pre-iL1B are upregulated in vaccinated individuals. Authors show that SYK signalling is activated in cells from vaccinated individuals, and they connect SYK activation with mtROS and NLRP3 activation. Co-cultivation experiments suggested that cells from vaccinated individuals drive the activation of effector memory T cells.

General comment.

The effect of Covid-19 vaccination on immune cells is certainly interesting. The work raises some interesting questions which may require follow up work to characterize in detail the underlying the mechanisms. The study is performed to a high standard using a relevant human model. I have the following comments for the authors' consideration.

1. Across the manuscript, authors show a number of immunoblots from patients' cells. It is not clear how many patients were probed. Standard practice is to test at least 5 different individuals. Therefore, in addition to the blots presents, authors should show as supplementary material the quantification of these blots using ImageJ plugs, for example. Without this additional information, these results cannot be considered reproducible.
2. Figure 1B to D. The most appropriate representation is to show in the same graph the unstimulated samples together with those challenged with S-protein and LPS. There should not be any problem combining the data in the same graph because these data should be obtained from the same individuals.
3. figure 2. Authors may wish to preSENT in greATER detail the RNA seq data.
4. There is an intriguing/puzzling observation that authors need to discuss/clarify. The RNA seq data convincingly show elevated levels of il1b in T2 cells. However somewhat surprisingly, LPS challenge of these cells did not result in increased levels of iL1B versus that produced by T0 cells. Is it possible that pre-iL1b protein levels do not correlate with the transcription data? Why is it not detected a priming effect for LPS?
5. Based on data shown in Fig 2E, authors infer that it is the S1 sub unit the one responsible for the phenotype. This can be experimental confirmed by challenging cells with recombinant S1 subunit (this can be requested to the authors of PMID: 34048708).
6. It is puzzling that LPS did not induce the phosphorylation of p65 in T0 cells. It is well establish that LPS does trigger p65 phosphorylation in human macrophages.
7. Authors have not disentangled whether the phenotype is due to increasing levels of pro-ILb, increasing activity of NLRP3, or both. A control will be to test whether SYK inhibitor affects the transcription of Il1b; and to test ASC speck formation in T2 cells

following treatment with the SYK inhibitor. The experiments done challenging THP-1 cells are not useful because these cells do not recapitulate the transcriptional changes observed in T2 cells.

8. A control to run is to inhibit mtROS, using mitoTEMPO for example, and then assessed whether IL1b levels are reduced in T2 cells.

9. Related to results shown in Fig 4. Are the results dependent on iLb, or are they inflammasome-dependent? Experiments probing IL1b blocking antibodies should help to clarify this issue.

10. Although the information is included in the figure legends, it is appropriate including in the statistical section of the methods the tests used.

Referee #2 (Remarks for Author):

The manuscript by Theobald et al. studies the innate-adaptive immune axis in response to SARS-CoV2 vaccination. The essential problem here is that it is known that an innate immune response is required for a productive adaptive immune response to SARS-CoV2, but with no (or minimal) adjuvant, how this is achieved is unclear. The authors utilize patients macrophages (differentiated from PBMCs) at various points of the vaccination process to study the innate immune arm of SARS-CoV2 vaccination.

The authors find that upon repeat vaccination, IL-1b release is increased and the inflammasome is activated in response to the product of the mRNA vaccine, S protein. The authors use RNAseq to trace this to mincle and CLEC4D. They then show that the tyrosine kinase Syk lies downstream of these C-type lectin receptors and mediates the signaling response. Lastly, the authors show that the Syk-inflammasome axis drives the trained innate immunity response.

The work is solid, well-triangulated and generally convincing. I have only small comments that should be easily addressable.

1. The S protein has been shown previously by this group and others to activate TLR2 (Nature Immunology, 22:829; eLife 2021;10:e68563, EMBO Molecular Medicine, 13:e14150). In Figure 1 of the current manuscript, however, at the T0 time, S protein cannot prime the macrophages. The implication is that without trained immunity the S protein is not recognized. Presumably, though, TLR2 is still there at T0. Do the authors have an explanation for the discordant results?

2. Along the lines of point 1, Figure 2 finds that mincle and CLEC4D are upregulated upon vaccination with the implication being that mincle and CLEC4D are responsible for the trained innate immunity. Again, how does TLR2 fit into this model?

3. It would be good to look at expression of mincle and CLEC4D in the samples in figure 1. Does the expression level match the responses? Additionally, figure 1 largely shows bimodal responses. Do the mincle and CLEC4D expression match the responses - that is, in high responders, are mincle and CLEC4D expression elevated?

4. Figure 3B should make more clear that these are stimulated cells. A western blot would also be helpful here, however, cell numbers may be an issue and if so, this would not be necessary in my mind.

5. In Figure 5K and 5L, how do you know the IFN is coming from the T cell? These are co-cultures with macrophages.

Minor points

1. The label "B" is missing in Figure 2B.

2. I find that line 122-124 (p.7) is overstated. Showing that an NF-kB inhibitor abrogates S-protein-mediated IL-1b secretion does not indicate that SYK promotes inflammasome formation through NF-kB signaling. This sentence should be re-worked.

In summary, this is a very strong manuscript that is well-triangulated and convincing. Attention to the logic and experimental details outlined above would make the manuscript even stronger.

Point-by-point response: Spleen tyrosine kinase mediates innate and adaptive immune crosstalk in SARS-CoV-2 mRNA vaccination; Theobald et al.

We thank the reviewer for the overall positive evaluation of our manuscript and their helpful comments. Before responding to the reviewers' comments point-by-point, we would like to give a general statement regarding our study cohort and samples used for our revision: Due to the success of the local vaccination programs as well as the high incidence of the Omicron-variant (B.1.1.529) of SARS-CoV-2 among the population, we were not able to recruit a sufficient number of unvaccinated/naïve individuals and individuals after the second vaccination. However, to answer the reviewers' questions and suggestions, we used macrophages isolated from individuals right before third vaccination and at least 6 months after 1st booster vaccination (2nd booster vaccination [T4]) as well as two weeks after this vaccination (T5). To confirm a similar reaction of the macrophages isolated at these time-points, we stimulated the cells with S-protein and nigericin and compared the results to our data from Fig. 1C. As expected T5 macrophages secreted IL-1b upon stimulation compared to T4 in comparable amount as T2 to T0. IL-1b release between T0 and T4 macrophages upon stimulation was equal (s. Revision Fig. 1). T4 and T5 were referred to as prior and post vaccination in the revised manuscript.

Revision Fig. 1: Monocytes were isolated prior (T0; n = 35) 1st vaccination and 2 weeks (T2; n = 44) after 2nd vaccination as well as before (T4; n = 12) and 2 weeks after (T5; n = 16) 3rd vaccination and differentiated to macrophages. IL-1β concentrations in supernatants of macrophages were measured by ELISA after stimulation with S-protein and nigericin as shown before. Box plots indicate the median and the upper and lower quartile. Outliers are plotted as individual dots (outside the 10-90 percentile). ***p<0.001.

Referee #1 (Comments on Novelty/Model System for Author):

The work was performed to a high quality standard for most of the experiments, including appropriate controls. Some of the findings reported (S protein being recognized by CLR receptors, SYK signalling mediating NLRP3 activation -though not shown in the context of SARS-CoV-2-) are published. However, the novelty arises from the use of relevant human samples with different vaccination status, indicating that vaccine primes only S protein-dependent NLRP3 activation. The work is of medical interest. The statistical tests are appropriate.

Referee #1 (Remarks for Author):

Theobald and colleagues investigate the effect of vaccination on inflammasome activation, and the effect on macrophage T- cell activation. Authors report that only cells from vaccinated individuals response to S-protein challenge releasing iL1B, and resulting in pyrptosis. RNA-seq data showed that the S-protein receptors and pre-iL1B are upreulated in vaccinated individuals. Authors show that SYK signalling is activated in cells from vaccinated individuals, and they connect SYK activation with mtROS and NLRP3 activation. Co-cultivation experiments suggested that cells from vaccinated individuals drive the activation of effector memory T cells.

General comment.

The effect of Covid-19 vaccination on immune cells is certainly interesting. The work raises some interesting questions which may require follow up work to characterize in detail the underlying the mechanisms. The study is performed to a high standard using a relevant human model. I have the following comments for the authors' consideration.

We thank the reviewer for the interest in our study and the positive comments.

1. Across the manuscript, authors show a number of immunoblots from patients' cells. It is not clear how many patients were probed. Standard practice is to test at least 5 different individuals. Therefore, in addition to the blots presents, authors should show as supplementary material the quantification of these blots using ImageJ plugs, for example. Without this additional information, these results cannot be considered reproducible.

We thank the reviewer for this comment and agree that a sufficient number of biological replicates is important. In our first draft, we used samples from three individuals, which were all represented as quantified graph in the expanded view. To support our hypothesis and represented data, we increased the number of biological replicates to 5 as suggested by the reviewer. All blots have been quantified using ImageJ and are depicted within the expanded view of the manuscript.

2. Figure 1B to D. The most appropriate representation is to show in the same graph the unstimulated samples together with those challenged with S-protein and LPS. There should not be any problem combining the data in the same graph because these data should be obtained from the same individuals.

We thank the reviewer for the comment and agree to the reviewer that this representation of the data is important. Thus, we have included the data in **Fig. EV1A**.

3. figure 2. Authors may wish to preSENT in greATER detail the RNA seq data.

To present our RNAseq data set in more detail we modified our volcano plot to increase the content of information. Moreover, we would like to refer to **Table EV1**: Here, we are providing a table with the DEGs with the log2fold change and the p-adjusted values of each gene. In addition, we now provide an accession number for the full dataset which was uploaded to GEO. We hope to meet the reviewer's suggestion.

*4. There is an intriguing/puzzling observation that authors need to discuss/clarify. The RNA seq data convincingly show elevated levels of *il1b* in T2 cells. However somewhat surprisingly, LPS challenge of these cells did not result in increased levels of *iL1B* versus that produced by T0 cells. Is it possible that pre-*iL1b* protein levels do not correlate with the transcription data? Why is it not detected a priming effect for LPS?*

We thank the reviewer for this important comment. In order to clarify the priming effect of LPS in T2 cells compared to T0 cells we have calculated the mean IL-1 β concentration of each timepoint upon LPS stimulation (see Revision Table 1) and blotted our data with a linear scale for a better visualization. As expected from the reviewer, a priming effect for LPS after vaccination can be observed clearly (see Revision Fig. 2). However, due to the high interindividual differences no significance could be overserved between the different timepoints.

Timepoint	T0	T1	T2	T3
N	35	28	44	33
Minimum IL-1 β concentration (pg/ml)	97	410	334	93
Maximum IL-1 β concentration (pg/ml)	13121	13561	13283	10837
Mean IL-1 β concentration (pg/ml)	3470	3540	4719	2984
SD +/-	3577	2777	3087	3151
SEM +/-	605	525	465	549

Revision Fig. 2: Monocytes were isolated prior (T0; n = 35) and 2 weeks after (T2; n = 28) 1st vaccination as well as 2 weeks (T2; n = 44) and 10 weeks (T3; n = 33) 2nd vaccination and differentiated to macrophages. IL-1 β concentrations in supernatants of macrophages were measured by ELISA after stimulation with S-protein. Box plots indicate the median and the upper and lower quartile. Outliers are plotted as individual dots (outside the 10-90 percentile).

Moreover, we would like to mention that a similar priming effect of LPS on *ex vivo* stimulated macrophages was observed in our previous study using healthy- and COVID-19 patient-derived macrophages (Theobald et al., 2021, EMBOMolMed). We assume that for LPS, the innate immune system is permanently triggered due to LPS producing bacteria which can be found as part of the normal skin and gut flora. The S-protein behaves differently and requires a priming step for full activation of the inflammasome. In our previous manuscript, we were able to link this effect to trained innate immunity in peripheral monocytes. In this manuscript we additionally identified SYK as a prerequisite for S-protein dependent signalling. This is now discussed in more detail.

5. Based on data shown in Fig 2E, authors infer that it is the S1 sub unit the one responsible for the phenotype. This can be experimental confirmed by challenging cells with recombinant S1 subunit (this can be requested to the authors of PMID: 34048708).

We thank reviewer 1 for this remark. To confirm our hypothesis that the S1 sub-domain is responsible for our observed effect, we stimulated macrophages isolated at timepoint T4 and T5 with the affinity purified S1 domain. Similar to our observation with the full-length protein (Revision Fig. 1), we observed a strong increase of the IL-1 β release at T5 upon S1 stimulation (new Fig. 2E) while stimulation with the subunit S2 led to no increase of the IL-1 β concentration (Fig. 2F). This observation is in line with our previous work (Theobald et al., 2021, EMBOMolMed) and supports the outstanding role of the S1 subunit in the S-protein-mediated inflammasome activation in macrophages.

6. It is puzzling that LPS did not induce the phosphorylation of p65 in T0 cells. It is well establish that LPS does trigger p65 phosphorylation in human macrophages.

We thank the reviewer for this important comment and agree that LPS does indeed trigger p65 phosphorylation in NF- κ B. To clarify this point we increased the number of biological replicates as illustrated in EV2C which now shows LPS triggered p65 phosphorylation. Moreover, we replaced the immunoblot of Fig. 3D for an unvaccinated individual with a more representative example.

7. Authors have not disentangled whether the phenotype is due to increasing levels of pro-ILb, increasing activity of NLRP3, or both. A control will be to test whether SYK inhibitor affects the transcription of Il1b; and to test ASC speck formation in T2 cells following treatment with the SYK inhibitor. The experiments done challenging tHP-1 cells are not useful because these cells do not recapitulate the transcriptional changes observed in T2 cells.

We thank the reviewer for this comment and agree that the suggested experiments are important to connect SYK signalling and inflammasome formation and activation. In order to meet the reviewer's comments we measured total IL-1b levels in the lysates derived from T0 and T2 or T5

macrophages. In-fact, we were able to observe an increase of total IL-1b in T2/T5 macrophages compared to T0 (see Revision Fig. 3).

Revision Fig. 3: Monocytes were isolated prior (n = 5) and after booster vaccination (n = 5). IL-1 β concentrations in unstimulated cell lysates of macrophages were measured by ELISA.

This observation is in line with our RNAseq data. Here we could detect a significant increase of the IL1B gene expression upon vaccination in unstimulated macrophages.

Gene	log2FoldChange	padjust
IL1B	3.736537136	0.002323004

We strongly believe that the observed priming of macrophages is dependent on multiple processes including an upregulation of the IL-1 β expression. As suggested by the reviewer and to finally link SYK signalling and inflammasome activation, we quantified ASC-SPECK formation in T5 macrophages untreated and treated with the SYK inhibitor (R406). As expected and in line with our THP-1 SYK knock-out line, ASC-SPECK formation was inhibited by the treatment with a SYK inhibitor (see new Fig. 3G). This finding also correlates with published data in which SYK is required for NLRP3 inflammasome assembly when cells are stimulated with alternative agents (PMID: 19339971; PMID: 25605870).

We strongly believe that the additional data now sufficiently link SYK signalling, NLRP3 inflammasome assembly and release of IL-1 β .

8. A control to run is to inhibit mtROS, using mitoTEMPO for example, and then assessed whether IL1b levels are reduced in T2 cells.

We thank for this excellent suggestion and agree that the usage of MitoTEMPO as additional control is important. In line with our hypothesis, treatment of T2 macrophages with MitoTEMPO decreased the IL-1 β secretion significantly as mitochondria play an important role in the pathway of spike-protein recognition and IL-1 β secretion (new Fig. EV2M)

9. Related to results shown in Fig 4. Are the results dependent on ILb, or are they inflammasome-dependent? Experiments probing IL1b blocking antibodies should help to clarify this issue.

We thank the reviewer for this important comment and agree that using an IL-1 β blocking antibody in the co-culture experiments is an important control. Thus, we performed experiments using an IL-1 β blocking antibody with T4 and T5 macrophages: Interestingly, the IL-1 β blocking antibody did not fully reverse the T cell phenotypes (less naive T cells and more effector memory T cells), as shown for R406 and MCC950 (New Figure EV4A-F). The same effects were observed, when we quantified the key T cell cytokines TNF α and IFN γ : Levels of these cytokines were reduced by an IL-1b blocking antibody but could not be fully abrogated (New Figure

EV4G,H). This might be explained by the fact that small molecules as R406 and MCC950 are fully blocking pyroptotic cell death of macrophages efficiently strongly decreasing release of DAMPs, which might also play a role in activation of T cells in co-culture experiments. However, usage of an IL-1 β blocking antibody showed a clear impact of T cell differentiation and activation indicating the outstanding role of IL-1 β in this process. This has now been discussed in the revised version of the manuscript.

10. Although the information is include din the figure legends, it is appropriate including in the statistical section of the methods the tests used.

All statistical methods used throughout the manuscript are now mentioned in the methods section.

Referee #2 (Remarks for Author):

The manuscript by Theobald et al. studies the innate-adaptive immune axis in response to SARS-CoV2 vaccination. The essential problem here is that it is known that an innate immune response is required for a productive adaptive immune response to SARS-CoV2, but with no (or minimal) adjuvant, how this is achieved is unclear. The authors utilize patients macrophages (differentiated from PBMCs) at various points of the vaccination process to study the innate immune arm of SARS-CoV2 vaccinaton.

The authors find that upon repeat vaccination, IL-1b release is increased and the inflammasome is activated in response to the product of the mRNA vaccine, S protein. The authors use RNAseq to trace this to mincle and CLEC4D. They then show that the tyrosine kinase Syk lies downstream of these C-type lectin receptors and mediates the signaling response. Lastly, the authors show that the Syk-inflammasome axis drives the trained innate immunity response.

The work is solid, well-triagulated and generally convincing. I have only small comments that should be easily addressable.

We are grateful for the positive evaluation of our manuscript by reviewer #2.

1. The S protein has been shown previously by this group and others to activate TLR2 (Nature Immunology, 22:829; eLife 2021;10:e68563, EMBO Molecular Medicine, 13:e14150). In Figure 1 of the current manuscript, however, at the T0 time, S protein cannot prime the macrophages. The implication is that without trained immunity the S protein is not recognized. Presumably, though, TLR2 is still there at T0. Do the authors have an explanation for the discordant results?

We agree with the author that this observation might be discordant as TLR2 should be also expressed on macrophages of unvaccinated individuals. To reveal the role of TLR2 in our model, we performed further experiments:

- 1.) We analysed the expression of TLR2 on macrophages prior and 2 weeks post-vaccination. Here, we could observe no increase of the TLR2 expression between the two groups quantified by flow cytometry. Which stands in contrast to the quantification of C-type lectins (see next point).

Revision Fig. 3: Monocytes were isolated prior (T4; n = 5) and 2 weeks after (post; n = 5) 3rd vaccination and differentiated to macrophages. TLR2 expression of macrophages was measured by flow cytometry. Graph indicate the mean fluorescence

intensity. Box plots indicate the median and the upper and lower quartile. All individual data points are plotted.

- 2.) We reanalysed our RNAseq data regarding the TLR2 expression: In line with the data from point 1.) we could not observe significant differences in T0 and T2 macrophages.

Gene	log2FoldChange	padjust
TLR2	0.114524687	0.287037936

- 3.) We additionally performed experiments where we used an anti-TLR2 blocking antibody on T5 macrophages, where we were able to see a reduction of the S protein induced IL-1b secretion, implicating that TLR2 is in-fact a contributing factor on in vivo primed macrophages. This confirms our recently published data (Theobald, EMBO MM, 2021)

Revision Fig. 4: Monocytes were isolated 2 weeks after (n = 5) 3rd vaccination and differentiated to macrophages. Prior treatment with S-protein cells have been incubated with a TLR2-blocking antibody in different concentrations. IL-1 β concentrations in supernatants of macrophages were measured by ELISA after stimulation with S-protein and nigericin. Box plots indicate the median and the upper and lower quartile. All individual data points are plotted.

In summary, our data implicate that TLR2 may play a role in the S-protein-mediated activation of the inflammasome and consecutive IL-1 β release. However, this observation cannot be explained by upregulation of TLR2 on primed macrophages. Given the fact that TLR2 is not upregulated following vaccination, we would prefer not to show the TLR2 data and rather focus on C-type lectins in this manuscript. Nevertheless, we discussed the possibility of the S-protein engaging several receptors including TLRs. See also comment below.

2. Along the lines of point 1, Figure 2 finds that mincle and CLEC4D are upregulated upon vaccination with the implication being that mincle and CLEC4D are responsible for the trained innate immunity. Again, how does TLR2 fit into this model?

We thank the reviewer for this comment and agree that the connection between CLECs and TLR-2 might be important. Interestingly, it is known that CLECs and TLR-2 can form hetero-complexes or signal dually upon binding of a ligand (PMID: 31867016). Within our current manuscript, we did not focus on this aspect, as the key point is the SYK signalling. However, we agree that this is an important point for future research. We added a paragraph to our discussion dealing with this aspect (line 279-282). As stated above, since we did not identify any upregulation of TLR2 in vaccinated individuals, we decided to focus on the CLEC/SYK signalling pathway and not to include speculations on the role of TLR2.

3. It would be good to look at expression of mincle and CLEC4D in the samples in figure 1. Does the expression level match the responses? Additionally, figure 1 largely shows bimodal responses. Do the mincle and CLEC4D expression match the responses - that is, in high responders, are mincle and CLEC4D expression elevated?

We agree that quantifying the expression levels on our macrophages at different time-points is an important experiment. Comparing T4 and T5 macrophages, we were able to detect significantly higher levels of mincle on T5 macrophages compared to T4. Upregulation of CLEC4D was borderline significant. These experiments confirm our RNAseq data, where both CLECs were higher expressed in T2 macrophages and additionally that mincle was higher expressed compared to CLEC4D. Overall, this new data fits well into our manuscript showing the importance of CLEC/SYK signalling for inflammasome activation and subsequent activation of an adaptive immune response. The new data has been added to Figure 2 (new G, H).

4. Figure 3B should make more clear that these are stimulated cells. A western blot would also be helpful here, however, cell numbers may be an issue and if so, this would not be necessary in my mind.

We thank the reviewer for this comment and for giving us the opportunity to clarify this. In fact, the macrophages used in this experiment were unstimulated. Unfortunately, this was described wrong in the materials and methods section and has been changed. Thus, vaccination primes phosphorylation of SYK which is then required for full inflammasome activation after re-stimulation.

5. In Figure 5K and 5L, how do you know the IFN is coming from the T cell? These are co-cultures with macrophages.

We agree that the measured IFN may potentially derive from both cell types in this co-culture assay. Nevertheless, at least to our knowledge, macrophages do not secrete relevant amounts of IFN upon any stimulation. Macrophages rather react to IFN secreted by T cells, since they naturally express IFN-receptors on their surface. Therefore, we strongly believe that the measured IFN is secreted by the T cells from this co-culture. This is also the principle of all commercially available IFN-release assays (e.g. Quantiferon) which measure T cell derived IFN following disease or vaccination.

Minor points

1. The label "B" is missing in Figure 2B.

This has been changed in the Figure.

2. I find that line 122-124 (p.7) is overstated. Showing that an NF-kB inhibitor abrogates S-protein-mediated IL-1b secretion does not indicate that SYK promotes inflammasome formation through NF-kB signaling. This sentence should be re-worked.

The sentence has been changed in the manuscript.

In summary, this is a very strong manuscript that is well-triangulated and convincing. Attention to the logic and experimental details outlined above would make the manuscript even stronger.

We like to thank the reviewer for the positive and helpful comments, helping us to improve our manuscript.

8th Jun 2022

Dear Dr. Rybniker,

Thank you for the submission of your revised manuscript to EMBO Molecular Medicine. I am pleased to inform you that we will be able to accept your manuscript pending the following final amendments:

- 1) Figures: Please upload individual, high-resolution figure files for EV Figures and add EV figure legends at the end of the main manuscript text. For more information on figure presentation please check "Author Guidelines".
<https://www.embopress.org/page/journal/17574684/authorguide#datapresentationformat>
- 2) In the main manuscript file, please do the following:
 - Correct/answer the track changes suggested by our data editors by working from the attached document.
 - Add up to 5 keywords.
 - Add callouts for Figure 1H.
 - Remove text highlight colour.
 - In M&M, statistical paragraph should reflect all information that you have filled in the Authors Checklist, especially regarding randomization, blinding, replication.
 - Please rename "Conflict of Interest" to "Disclosure Statement & Competing Interests". We updated our journal's competing interests policy in January 2022 and request authors to consider both actual and perceived competing interests. Please review the policy <https://www.embopress.org/competing-interests> and update your competing interests if necessary.
 - In addition to the accession number please provide URL all deposited datasets. Use the following format to report the accession number of your data:

[data type]: [full name of the resource] [accession number/identifier] ([doi or URL or identifiers.org/DATABASE:ACCESSION])

Please check "Author Guidelines" for more information.

<https://www.embopress.org/page/journal/17574684/authorguide#availabilityofpublishedmaterial>

3) Funding: Please make sure that information about all sources of funding are complete in both our submission system and in the manuscript.

4) Author Checklist: We updated the "Author Checklist". Please complete the new version and submit it with the revised manuscript. You can find it here: <https://www.embopress.org/page/journal/17574684/authorguide>

5) Author contributions: Please specify author contributions in our submission system. CRediT has replaced the traditional author contributions section because it offers a systematic machine readable author contributions format that allows for more effective research assessment. You are encouraged to use the free text boxes beneath each contributing author's name to add specific details on the author's contribution. More information is available in our guide to authors:

<https://www.embopress.org/page/journal/17574684/authorguide#authorshippinguidelines>

6) Synopsis: Every published paper now includes a 'Synopsis' to further enhance discoverability. Synopses are displayed on the journal webpage and are freely accessible to all readers. They include separate synopsis image and synopsis text.

- Synopsis image: Please annotate the provided image (e.g. which cells, blue is DAPI but what is marked by green colour) or provide a visual abstract as a high-resolution jpeg file 550 px-wide x (250-400)-px high to illustrate your article.

- Synopsis text: Please provide a short standfirst (maximum of 300 characters, including space) as well as 2-5 one sentence bullet points that summarise the paper as a .doc file. Please write the bullet points to summarise the key NEW findings. They should be designed to be complementary to the abstract - i.e. not repeat the same text. We encourage inclusion of key acronyms and quantitative information (maximum of 30 words / bullet point). Please use the passive voice.

7) For more information: Please remove corresponding author's e-mail address. This space should be used to list relevant web links for further consultation by our readers. Could you identify some relevant ones and provide such information as well? Some examples are patient associations, relevant databases, OMIM/proteins/genes links, author's websites, etc...

8) Source data: Please label all source data and submit one file per figure for the main figures and a zipp file for the EV figures. Please check "Author Guidelines" for more information.

<https://www.embopress.org/page/journal/17574684/authorguide#sourcedata>

9) As part of the EMBO Publications transparent editorial process initiative (see our Editorial at <http://embomolmed.embopress.org/content/2/9/329>), EMBO Molecular Medicine will publish online a Review Process File (RPF) to accompany accepted manuscripts. This file will be published in conjunction with your paper and will include the anonymous referee reports, your point-by-point response and all pertinent correspondence relating to the manuscript. Let us know whether you agree with the publication of the RPF and as here, if you want to remove or not any figures from it prior to publication. Please note that the Authors checklist will be published at the end of the RPF.

10) Please provide a point-by-point letter INCLUDING my comments as well as the reviewer's reports and your detailed responses (as Word file).

I look forward to reading a new revised version of your manuscript as soon as possible.

Yours sincerely,

Zeljko Durdevic

2nd Authors' Response to Reviewers and Editor

We thank the reviewers for the final positive evaluation of our manuscript and for helping us to strengthen our data.

All points made by the Editor for the final amendments were addressed in our revised manuscript.

We are pleased to inform you that your manuscript is accepted for publication and is now being sent to our publisher to be included in the next available issue of EMBO Molecular Medicine.